# Rendezvous in Cis-Lunar Space near Rectilinear Halo Orbit: Dynamics and Control Issues

Giordana Bucchioni [†] and Mario Innocenti *,[†]

Department of Information Engineering, University of Pisa, 56126 Pisa, Italy; giordana.bucchioni@hotmail.it
* Correspondence: mario.innocenti@unipi.it
† These authors contributed equally to this work.

**Abstract:** The paper presents the development of a fully-safe, automatic rendezvous strategy between a passive vehicle and an active one orbiting around the Earth–Moon L2 Lagrangian point. This is one of the critical phases of future missions to permanently return to the Moon, which are of interest to the majority of space organizations. The first step in the study is the derivation of a suitable full 6-DOF relative motion model in the Local Vertical Local Horizontal reference frame, most suitable for the design of the guidance. The main dynamic model is approximated using both the elliptic and circular three-body motion, due to the contribution of Earth and Moon gravity. A rather detailed set of sensors and actuator dynamics was also implemented in order to ensure the reliability of the guidance algorithms. The selection of guidance and control is presented, and evaluated using a sample scenario as described by ESA's HERACLES program. The safety, in particular the passive safety, concept is introduced and different techniques to guarantee it are discussed that exploit the ideas of stable and unstable manifolds to intrinsically guarantee some properties at each hold-point, in which the rendezvous trajectory is divided. Finally, the rendezvous dynamics are validated using available Ephemeris models in order to verify the validity of the results and their limitations for future more detailed design.

**Keywords:** rendezvous; NRHO; three-body problem

## 1. Introduction

The paper presents some of the dynamics models and potential guidance algorithms usable for the preliminary assessment of a rendezvous between an unmanned vehicle ascending from the Moon, and a permanent manned/unmanned service station located in a collinear Lagrangian point orbit of the Earth–Moon system. The motivation for this work is given by the renewed interest of many government agencies and private organizations in returning to our satellite.

The positive aspects of one such parking orbit lie in the possibility of continuous communications with Earth, the visibility of the lunar hidden surface, and the relative low cost of station keeping maneuvers. Lagrangian points are known to be equilibria within a circular restricted three-body model. Although the model is an approximation of the real multibody dynamics, it serves for a useful series of analyses, in particular when it comes to guidance and control algorithms, which, by nature, need simplified modeling for their primary design.

The attractiveness of the above orbits (Lissajous, halo, etc.) has been exploited over the years by several missions, especially in the Sun–Earth system. The Solar and Heliospheric Observatory and the Deep Space Climate Observatory are located at the L1 collinear point, while the Wilkinson Microwave Anisotropy Probe and the Gaia spacecraft used the L2 collinear point. The orbit of the James Webb telescope will be also located at the latter.

While the previous examples focused on science experiments and scientific knowledge, the use of Lagrangian points in the Earth–Moon system is also considered for future human

exploration. Of particular interest, we can mention the Chinese Chang'e series missions, especially the number four, with the presence of the relay satellite Queqiao. The spacecraft took 24 days to reach L2 from Earth, using a lunar swing-by to save fuel. On 14 June 2018, Queqiao finished its final adjustment burn and entered the L2 halo mission orbit, which is about 65,000 kilometers from the Moon. This is the first lunar relay satellite at that location. The ARTEMIS mission instead is a programmed multi-agency mission with the objective of a permanent return to the Moon inclusive of human presence. Within ARTEMIS, a permanent station in an L2 halo orbit will serve as a bridge between human/robotic activities on the Moon, and the transfer of astronauts and material to and from the Earth and the Moon.

The present paper is cast within NASA's ARTEMIS program and considers one specific aspect of ESA's Heracles mission concept [1], which is the relative dynamics and guidance issues of the automated rendezvous between a lunar ascender vehicle and an L2 orbiting station.

Since this type of mission has not been performed yet, the paper presents several aspects of originality. The first aspect is relative to the dynamic modeling. It is important to establish the level of modeling necessary to describe the relative motion between the chasing vehicle and the target station. Keeping in mind that we are interested in a preliminary model for guidance and control, the paper describes a comparison between elliptic and circular three-body motions and validates their propagation against an Ephemeris model using GMAT and JPL ephemerides data. The inclusion of the Sun as a fourth body is considered negligible, and only potential effects of the solar radiation pressure are investigated. The incorporation of a set of sensors and actuators is also considered, in order to verify propagation errors due to the presence of their models in the dynamics. Another contribution is the selection of a set of guidance algorithms for open loop and closed loop control of the chaser, depending on the distance from the target. Guidance methods are not proposed as the optimal ones; instead, they are considered as "potential" feasibility solutions. Finally, numerical simulations are performed using the scenario defined by [1] as an example.

The background mission is described in Section 2: the mathematics relative to relative motion and their validation are described in Sections 3 and 4; Section 5 describes the models used for the primary sensors and actuators; Section 6 presents the structure of the guidance algorithms, whose details are reported in the appendix. Finally, numerical results and conclusions are shown in Sections 7 and 8.

## 2. Background

This section describes the general sequence of phases within ESA's Heracles (currently renamed ESA Large Logistic Lander or EL3) mission. Figures 1 and 2 show an artistic representation of the lunar ascent vehicle, which would rendezvous with a station in orbit.

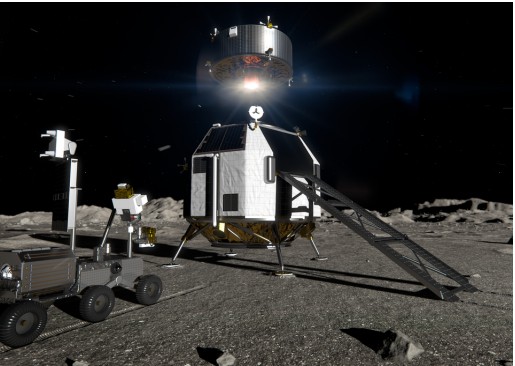

**Figure 1.** Lunar lander structure.

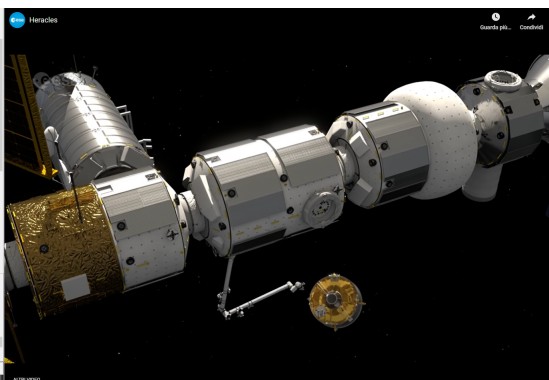

**Figure 2.** Rendezvous and berthing.

The Heracles project was initially planned with the objective of delivering Moon samples to Earth on NASA's Orion spacecraft as early as its fourth or fifth mission. The initial project was set up to be integrated with NASA's plan to return to the Moon, and to build a permanent station (Gateway) in cislunar orbit. This has now become NASA's Artemis mission.

A small lander with a rover inside weighing around 1800 kg in total would land and be monitored by astronauts from the Gateway. The current status of the mission includes two distinct tasks: the first is to deploy a rover whose objective is the collection of soil samples, the second is to launch a module that will carry samples to the orbiting station. When the ascent module carrying the sample container arrives, the Gateway's robotic arm will capture it and berth it with the outpost's airlock for unpacking and transfer of the container to Orion and subsequent unmanned flights to Earth and later on with returning astronauts.

The aim of the mission is to prove the advancements in autonomous operations in space, in particular the focus of this paper is on the sequence of phasing and rendezvous of the LAE with the orbiting LOP-G. The first part of the mission starts with the phasing maneuver that sees the transferring of the LAE-also called chaser in the following—with the payload of lunar soil, collected from the south pole of the Moon, from an assumed low lunar orbit with an altitude of 100 km towards the L2 Near Rectilinear Halo Orbit (NRHO) in which the LOP-G—also called target—is orbiting. The rendezvous maneuver will take care of the approach when the relative distance between the two space vehicles is less than 100 km.

The literature and the technical details will be discussed in the dedicated sections, and the goal of this section is instead to provide a general background for of the challenges that the design of a transfer maneuver in such peculiar environment implies.

One of the critical aspects of this problem is the formulation of the proper equations of motion that should be written in non-inertial reference frame and must describe the complete 6-DOF relative dynamics for the rendezvous and berthing; however, under the restricted three-body problem hypothesis, there is no closed form solution and no standard procedure to compute the best rendezvous strategy and to select the best guidance/control technique.

Phasing maneuvers are continuously performed in low Earth orbit to connect to the ISS, but no experience is available in the case of automated motion in the restricted three-body problem. The LAE must fly towards the Gateway departing from a Low Lunar Orbit to a L2-NRHO, minimizing the fuel consumption and the required time of flight. The degrees of freedom of the maneuver are multiple. A possibility is to take advantage of the manifold theory, with the dynamics being modeled under the circular restricted three-body problem, in order to have a preliminary evaluation of the influence of design parameters such as TOF, number of impulses, and their amplitude.

Once the phasing is accomplished, the rendezvous phase may start immediately or after one orbit or two orbits depending on the acquired relative initial conditions. The rendezvous is a critical phase of the approaching strategy, since the two vehicles are very

close to one another and the risk of collision or mission abortion is critically high. The standard closing maneuver usually follows a series of way points located in which the chaser must stop and check its state and the relative state according to accuracy, changes to sensor suite, and safety. During the rendezvous, multiple changes of the types of sensors, guidance, and control methods may be needed; therefore, all of this instrumentation must be properly selected, modeled, and validated via simulation. One of the main requirements, for instance, is the use of cameras for information on translational as well as attitude state variables.

As an example, Figure 3 shows a conceptual scheme of the sequence of the complete maneuver for the mission under study in the paper.

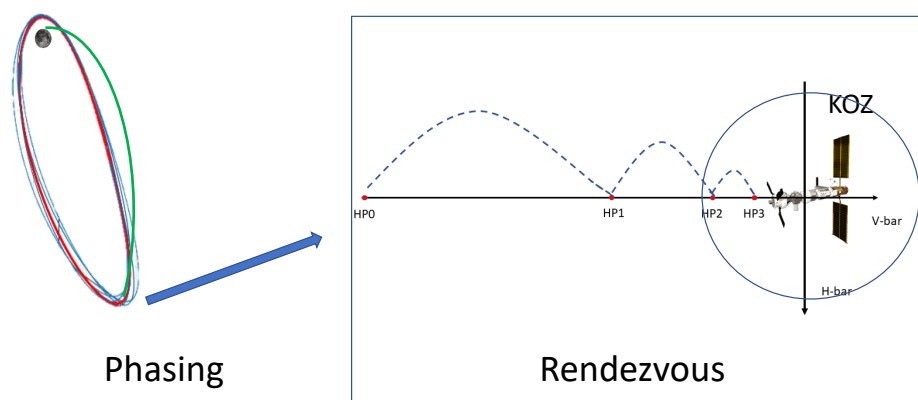

**Figure 3.** Phasing and rendezvous maneuver example.

## 3. Relative Dynamics Equations of Motion

This section reviews the relative motion of the two spacecraft within the Earth–Moon system, or, more generally, the motion of three bodies under mutual gravitational attraction. The design of the rendezvous and docking maneuvers requires an accurate formulation of the relative motion; in this work, some working hypotheses are made, such as the elliptic and circular restricted three-body problem hypotheses. In addition, the equations of motions are formulated in non-inertial frames to make them more useful in the analysis and synthesis of the GNC loop. The complexity of this formulation stays in the fact that no closed solution exists for the problem. The interested reader can refer to [2] for more details. The appropriate reference frames are reviewed herein; then, translational and rotational dynamics are described.

### 3.1. Reference Frames

The generic *inertial frame* centered in $O$ and with unit vectors $\hat{I}$, $\hat{J}$, and $\hat{K}$, will be denoted as follows:

$$\mathcal{I}: \left\{ C; \hat{I}, \hat{J}, \hat{K} \right\} \tag{1}$$

Consider two primary bodies, with masses $M_1$ and $M_2$, orbiting around their composite center of mass $C$ in a collinear formation. A convenient frame for describing the motion of spacecraft in such a system is the *synodic reference frame*. It can be centered in one of the primaries center of mass, as in [3], and the unit vectors are defined as follows:

- $\hat{i}_s = -r_{12}/\|r_{12}\|$ where $r_{12}$ is the position of $M_2$ with respect to $M_1$;
- $\hat{k}_s$ is perpendicular to the plane where the primaries revolve, and is positive in the direction of the system angular velocity vector;
- $\hat{j}_s = \hat{k}_s \times \hat{i}_s$ completes the right-handed coordinate systems.

This choice may be convenient when spacecraft measurements are taken with respect to the nearest primary. For instance, in [3], the Sun–Earth/Moon system is considered, and the synodic frame is centered on the Earth, since the spacecraft receives measurements with respect to it. In the following, we will refer to such a coordinate system as *primary-centered rotating frame*, and it will be denoted as

$$\mathcal{M}_i : \left\{ O_i; \hat{\imath}_m, \hat{\jmath}_m, \hat{k}_m \right\} \tag{2}$$

where *i* is the index of the primary chosen for placing the coordinate frame. Figure 4 shows the primary-centered rotating frame, centered on $M_2$, and the Local-Vertical Local Horizon Frame, described next.

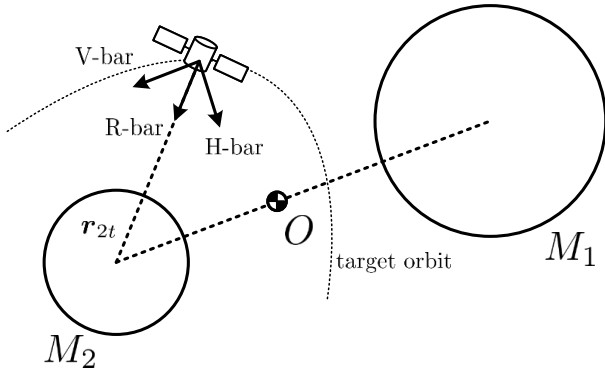

(**a**) Local-vertical local horizon frame $\mathcal{L}_2$: in the picture, it is possible to see a schematic representation of the LVLH frame

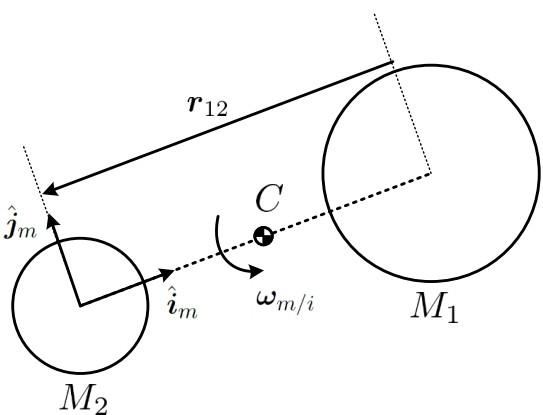

(**b**) Primary-centered rotating reference frame $\mathcal{M}_2$

**Figure 4.** Reference frames.

Rendezvous trajectories are generally described in a frame local to the target. This eases the analysis and the trajectory monitoring of incoming vehicles, as well as the definition of keep-out zones and admissible approaching corridors. The LVLH frame is usually employed for rendezvous scenario analysis, and defined as:

$$\mathcal{L}_i : \left\{ r_{it}; \hat{\imath}, \hat{\jmath}, \hat{k} \right\} \tag{3}$$

The LVLH (Sometimes called orbital frame) frame is defined with respect to the primary body around which the target is orbiting. Denoting with $r_{it}$ the target position with respect to the primary $i$, with $\left[\dot{r}_{it}\right]_{\mathcal{M}_i}$, the target velocity as seen from the primary, and with $h_{it} = r_{it} \times \left[\dot{r}_{it}\right]_{\mathcal{M}_i}$, the target specific angular momentum with respect to the primary, the LVLH frame unit vectors are defined and named as follows:

- $\hat{k} = -r_{it}/\|r_{it}\|$ points to the primary and is called *R-bar*;
- $\hat{j} = -h_{it}/\|h_{it}\|$ is perpendicular to the target instantaneous orbital plane (negative specific angular momentum direction) and is called *H-bar*;
- $\hat{i} = \hat{j} \times \hat{k}$ completes the right-handed reference frame, and is called *V-bar*.

The above definition of the LVLH frame is consistent with the one given by Fehse in its classical reference book for spacecraft rendezvous and docking [4] (pp. 31–32, Section 3.1.3). The LVLH frame for a target orbiting around $M_2$ is shown in Figure 4.

Note that the formal definition of the LVLH frame is relative to Kaplerian motion so using R-bar, H-bar, and V-bar is somewhat an abuse of notation, justified by their use in the community. In addition to the above, other body fixed reference frames can be used to highlight the dynamics relative to specific points of the vehicles (docking port, geometric center of mass, location of actuators, etc.).

### 3.2. Translational Equations of Relative Motion

Consider a *target* and a *chaser* spacecraft, orbiting around the Moon, and subjected to both Earth and Moon gravitational influence. Their equations of motion with respect to the Moon are:

$$\left[\ddot{r}\right]_{\mathcal{I}} = -\mu \frac{r}{r^3} - (1-\mu)\left(\frac{r + r_{em}}{\|r + r_{em}\|^3} - \frac{r_{em}}{r_{em}^3}\right) \tag{4}$$

$$\left[\ddot{r}_c\right]_{\mathcal{I}} = -\mu \frac{r_c}{r_c^3} - (1-\mu)\left(\frac{r_c + r_{em}}{\|r_c + r_{em}\|^3} - \frac{r_{em}}{r_{em}^3}\right) \tag{5}$$

where:

- $\mu = 1.215 \times 10^{-2}$ is the Moon–Earth mass ratio parameter defined as:
  $\mu = \frac{\mu_m}{\mu_e + \mu_m} = (1 + \frac{M_e}{M_m})^{-1}$ with $M_e + M_m = 1$
- $r_{em}$ is the position of the Moon with respect to the Earth, and $r_{em} = \|r_{em}\|$ its norm;
- $r$ and $r_c$ are the target and chaser positions with respect to the Moon, with norms $r = \|r\|$, and $r_c = \|r_c\|$, respectively.

In order to simplify computation, distances are normalized to the Moon semi-major axis and time to the Moon mean angular motion.

With reference to Figure 5, the chaser position with respect to the Moon is given by:

$$r_c = r + \rho \tag{6}$$

where $\rho$ is the relative position of the chaser with respect to the target. Taking the second time derivative of Equation (6) and using Equations (4) and (5), we obtain the nonlinear equations of relative motion in the LVLH reference frame:

$$\left[\ddot{\rho}\right]_{\mathcal{L}} + 2\omega_{l/i} \times \left[\dot{\rho}\right]_{\mathcal{L}} + \left[\dot{\omega}_{l/i}\right]_{\mathcal{L}} \times \rho + \omega_{l/i} \times (\omega_{l/i} \times \rho)$$
$$= \mu\left(\frac{r}{r^3} - \frac{r + \rho}{\|r + \rho\|^3}\right) + (1-\mu)\left(\frac{r + r_{em}}{\|r + r_{em}\|^3} - \frac{r + \rho + r_{em}}{\|r + \rho + r_{em}\|^3}\right) \tag{7}$$

Recall that:

$$\left[\dot{\omega}_{l/i}\right]_{\mathcal{I}} = \left[\dot{\omega}_{l/i}\right]_{\mathcal{L}} \tag{8}$$

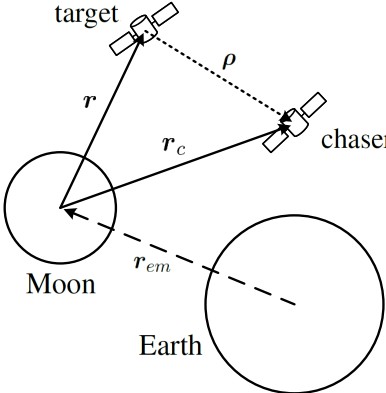

**Figure 5.** Target and chaser spacecraft in the three-body system.

Using the subscript notation for the reference frames above, the angular velocity of the LVLH frame with respect to the inertial frame can be computed by simple composition:

$$\boldsymbol{\omega}_{l/i} = \boldsymbol{\omega}_{l/m} + \boldsymbol{\omega}_{m/i} \tag{9}$$

where $\boldsymbol{\omega}_{l/m}$ and $\boldsymbol{\omega}_{m/i}$ are the angular velocities of $\mathcal{L}$ with respect to $\mathcal{M}$, and of $\mathcal{M}$ with respect to $\mathcal{I}$, respectively. The LVLH angular acceleration with respect to $\mathcal{I}$ is given by:

$$\left[\dot{\boldsymbol{\omega}}_{l/i}\right]_{\mathcal{L}} = \left[\dot{\boldsymbol{\omega}}_{l/m}\right]_{\mathcal{L}} + \left[\dot{\boldsymbol{\omega}}_{m/i}\right]_{\mathcal{L}} = \left[\dot{\boldsymbol{\omega}}_{l/m}\right]_{\mathcal{L}} + \left[\dot{\boldsymbol{\omega}}_{m/i}\right]_{\mathcal{M}} - \boldsymbol{\omega}_{l/m} \times \boldsymbol{\omega}_{m/i} \tag{10}$$

Equation (7), along with Equations (9) and (10), constitutes a set of time-varying nonlinear equations:

- $r$, $\boldsymbol{\omega}_{l/m}$, and $\left[\dot{\boldsymbol{\omega}}_{l/m}\right]_{\mathcal{L}}$ that depend on the target motion around the Moon;
- $r_{em}$, $\boldsymbol{\omega}_{m/i}$, and $\left[\dot{\boldsymbol{\omega}}_{m/i}\right]_{\mathcal{M}}$, characteristics of the Moon orbital motion.

Equation (7) can be further simplified if we consider the restricted three-body problem (elliptic or circular), which will be actually used later on in the paper for the guidance design.

### 3.3. Attitude Equations of Motion

Traditionally, orbital mechanics refers to trajectory dynamics; however, in our problem, attitude motion becomes especially critical in order to maintain correct orientation in the final phase (berthing or docking).

Standard derivation using rigid body approximation uses Euler's rotation equations with the general form [5]:

$$\mathbb{I}\dot{\boldsymbol{\omega}} + \boldsymbol{\omega} \times \mathbb{I}\boldsymbol{\omega} = \mathbf{N} \tag{11}$$

where $\mathbf{N}$ is the vector of external torques, $\mathbb{I}$ is the inertia tensor matrix, and $\boldsymbol{\omega}$ is the angular velocity about (typically) the principal axes.

As angular velocity vectors are cumulative, we can write the inertial angular velocity as:

$$\boldsymbol{\omega} = \boldsymbol{\omega}_{co} + \boldsymbol{\omega}_{oi}$$

where $\boldsymbol{\omega}$ is the angular rate of body chaser frame w.r.t. inertial frame, $\boldsymbol{\omega}_{co}$ (or $\boldsymbol{\omega}_c$) is the angular rate of body chaser frame w.r.t. orbital frame and $\boldsymbol{\omega}_{oi}$ is the angular velocity of orbital frame w.r.t. inertial frame. Note that all vectors are expressed in chaser body frame. For the kinematics, we derive the differential equations of motion of the body frame w.r.t. the LVLH reference frame, relating the Euler angles (in their Bryant 3, 2, 1 sequence) with the angular velocity vector $\boldsymbol{\omega}_c$ [6,7]. Using the standard Euler matrix rotation, we can write $\boldsymbol{\omega}_c$ as a function of Euler angles. The inverse relationship becomes

$$\begin{bmatrix} \dot{\theta}_x \\ \dot{\theta}_y \\ \dot{\theta}_z \end{bmatrix} = \frac{1}{\cos(\theta_y)} \begin{bmatrix} \cos(\theta_y) & \sin(\theta_x)\sin(\theta_y) & \cos(\theta_x)\sin(\theta_y) \\ 0 & \cos(\theta_x)\cos(\theta_y) & -\sin(\theta_x)\cos(\theta_y) \\ 0 & \sin(\theta_x) & \cos(\theta_x) \end{bmatrix} \boldsymbol{\omega}_c = \mathbf{J}(\boldsymbol{\theta}_c)^{-1}\boldsymbol{\omega}_c \quad (12)$$

Quaternions can also be used if singularities are critical. The interested reader can refer to [5,6], for instance.

The target's attitude model is taken from the literature to be a saw tooth type of motion, with an amplitude of one degree and a frequency of 1 Hz [4,7].

### 3.4. Attitude Relative Motion

The relative attitude between the two spacecraft is based on the angular rate vectors, since Euler angles are not cumulative. Thus:

$$\boldsymbol{\omega}_{ra} = \boldsymbol{\omega}_c - \mathbf{R}_{co}(\boldsymbol{\theta}_c)\boldsymbol{\omega}_t \quad (13)$$

where $\boldsymbol{\omega}_{ra}$ is the relative angular rate in chaser body frame, $\boldsymbol{\omega}_c$ is the chaser angular rate expressed in chaser frame, $\boldsymbol{\omega}_t$ is the target angular rate expressed in orbital frame and $\mathbf{R}_{co}$ is the rotation matrix that transforms a vector from orbital frame components to body chaser frame components

$$\mathbf{R}_{co}(\boldsymbol{\theta}_c) = \begin{bmatrix} 1 & 0 & 0 \\ 0 & \cos(\theta_{c_x}) & \sin(\theta_{c_x}) \\ 0 & -\sin(\theta_{c_x}) & \cos(\theta_{c_x}) \end{bmatrix} \begin{bmatrix} \cos(\theta_{c_y}) & 0 & -\sin(\theta_{c_y}) \\ 0 & 1 & 0 \\ \sin(\theta_{c_y}) & 0 & \cos(\theta_{c_y}) \end{bmatrix} \begin{bmatrix} \cos(\theta_{c_z}) & \sin(\theta_{c_z}) & 0 \\ -\sin(\theta_{c_z}) & \cos(\theta_{c_z}) & 0 \\ 0 & 0 & 1 \end{bmatrix} \quad (14)$$

Using Equation (13), we can express the derivative of the relative Euler angles $\dot{\boldsymbol{\theta}}_{ra}$ as follows:

$$\dot{\boldsymbol{\theta}}_{ra} = \mathbf{J}(\boldsymbol{\theta}_{ra})^{-1}\boldsymbol{\omega}_c - \mathbf{J}(\boldsymbol{\theta}_{ra})^{-1}\mathbf{R}_{co}(\boldsymbol{\theta}_c)\boldsymbol{\omega}_t \quad (15)$$

where $\mathbf{J}(\boldsymbol{\theta}_{ra})$ is the appropriate Jacobian matrix.

For close proximity maneuvers, we need the relative centers of mass position of interest, and the position and velocity between two docking ports, which are located elsewhere on the spacecraft. The port to port distance can be expressed as

$$\boldsymbol{\rho}_{pp} = \boldsymbol{\rho} + \mathbf{r}_{dc} - \mathbf{r}_{dt} \quad (16)$$

where $\boldsymbol{\rho}$ is the relative distance between chaser and target CoM, $\mathbf{r}_{dc}$ and $\mathbf{r}_{dt}$ are the docking port position of chaser and target, respectively. The equation above can be seen as an output of the system.

The target (or chaser) docking port in the orbital reference frame LVLH can be presented as

$$\begin{bmatrix} \mathbf{r}_{pi} \end{bmatrix}_o = \mathbf{R}_{io}(\boldsymbol{\theta}_i)^T \begin{bmatrix} \mathbf{r}_{pi} \end{bmatrix}_i \quad (17)$$

with the rotation matrix defined as follows:

$$\mathbf{R}_{io}(\boldsymbol{\theta}_i) = \begin{bmatrix} 1 & 0 & 0 \\ 0 & \cos(\theta_{i_x}) & \sin(\theta_{i_x}) \\ 0 & -\sin(\theta_{i_x}) & \cos(\theta_{i_x}) \end{bmatrix} \begin{bmatrix} \cos(\theta_{i_y}) & 0 & -\sin(\theta_{i_y}) \\ 0 & 1 & 0 \\ \sin(\theta_{i_y}) & 0 & \cos(\theta_{i_y}) \end{bmatrix} \begin{bmatrix} \cos(\theta_{i_z}) & \sin(\theta_{i_z}) & 0 \\ -\sin(\theta_{i_z}) & \cos(\theta_{i_z}) & 0 \\ 0 & 0 & 1 \end{bmatrix} \quad (18)$$

The subscript $i = c, t$ indicate chaser or target, respectively.

## 4. Validation of the Equations of Motion

This section describes in detail the validation process performed by comparing the model of the previous section with available Ephemeris models. The dynamics of the target

orbit, relative ER3BP, and CR3BP motions are evaluated for a free drift motion and the most significant rendezvous scenario.

The Ephemeris model was not directly implemented and the validated commercial software GMAT was selected [8], since it could be easily integrated with the propagation simulator. The set DE405 generated by JPL was used in the comparison.

### 4.1. Propagation of the Equations of Motion

This section presents the propagation of equations of motion over one target orbit with no active corrections. The comparison was performed in the following way:
Elliptic restricted, circular restricted, and GMAT were initialized with the same initial conditions in terms of position and velocity in the synodic reference frame $\mathcal{S}$, and the dynamics were propagated for one orbit. The same procedure was performed for 100 times at different Mean Anomalies ($M$), from 0 to 180 degrees. The one orbit propagation considered to account for a second rendezvous was necessary if the first was not successful. GMAT was set to simulate the restricted three-body problem dynamics as well.

The orbit propagation in the three different models (GMAT, ER3BP, CR3BP) is shown in Figure 6; in each sub figure, it is possible to see all the trajectories propagated for the duration of seven days. The propagation duration is taken as representative of the orbital period of the target vehicle; even though the Ephemeris propagation is more realistic, the obtained trajectories require some active corrections to be considered orbits. The starting point at the periselene propagation is in blue, whereas, at the aposelene, it is in red.

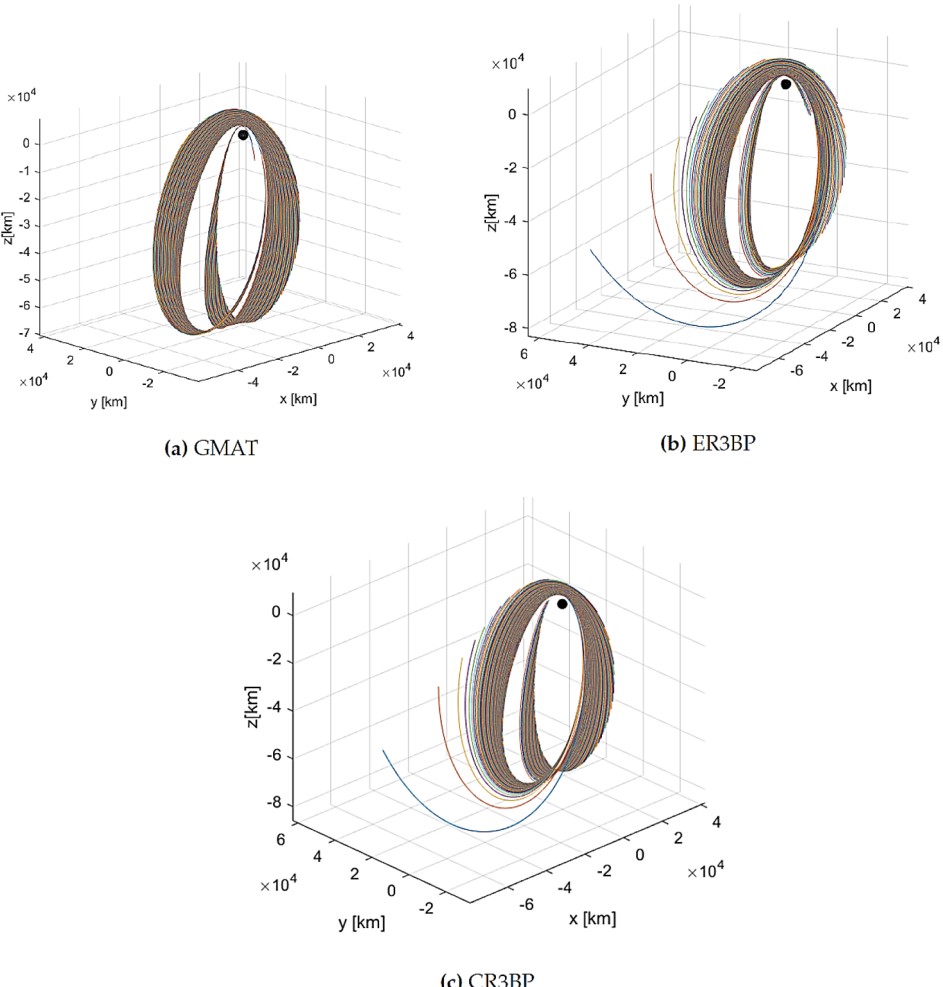

**(a)** GMAT         **(b)** ER3BP

**(c)** CR3BP

**Figure 6.** Propagated trajectories.

### 4.1.1. Comparison Results

The next figures and present the comparison between ER3BP, CR3BP, and GMAT. Figure 7 shows the position (top column) and velocity (bottom column) differences between ER3BP and GMAT for one orbit period and the details at the aposelene. The numerical maximum and minimum values are reported in Tables 1 and 2. Note that the aposelene region is the one that is assumed to provide better rendezvous properties. The position error increases with the simulation time, and it increases more rapidly as the starting point is closer to the periselene. This latter result is expected since the unmodelled dynamics play a larger role.

Analogous tests were performed using the CR3BP and shown in Figure 8 by using the same simulation procedure. It is possible to observe that, as expected, the CR3BP model has larger errors with respect to the Ephemeris than the ER3BP model. Moreover, the error amplitude can be tolerated for mean anomalies between 100 and 180 degrees. The numerical errors in Tables 1 and 2 indicate the values propagated trajectory with respect to GMAT. The tables highlight how the propagation errors increase with the three-body restricted models with respect to the Ephemerides.

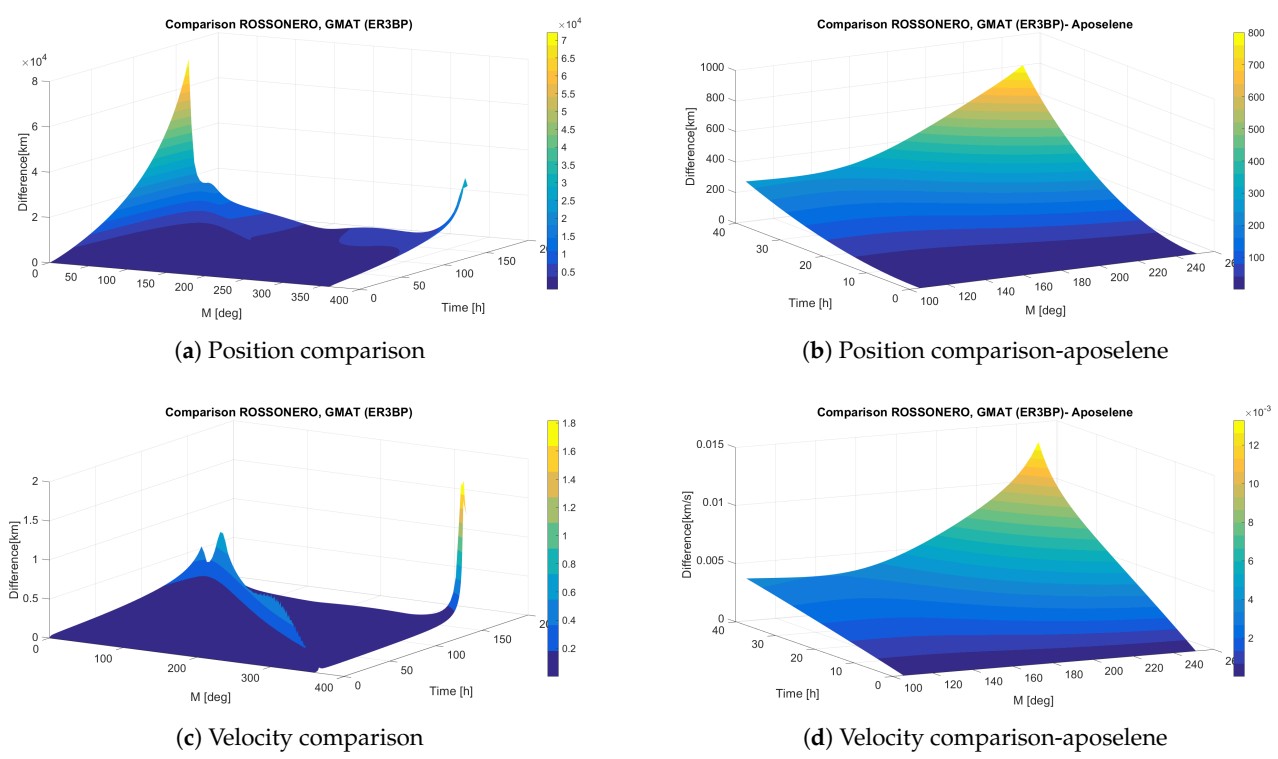

(**a**) Position comparison      (**b**) Position comparison-aposelene

(**c**) Velocity comparison      (**d**) Velocity comparison-aposelene

**Figure 7.** ER3BP and Ephemeris comparison.

**Table 1.** Maximum propagation errors.

|  | **ER3BP** | **CR3BP** |
| --- | --- | --- |
| Position Error | $7.2 \times 10^4$ km | $7.35 \times 10^4$ km |
| Velocity Error | 1.8 km/s | 1.8 km/s |
| Position Error-Aposelene | 800 km | 556 km |
| Velocity Error-Aposelene | 0.01 km/s | 0.009 km/s |

**Table 2.** Minimum propagation errors.

|  | **ER3BP** | **CR3BP** |
|---|---|---|
| Position Error | $2.3 \times 10^4$ km | $1.755 \times 10^4$ km |
| Velocity Error | 0.006 km/s | 0.003 km/s |
| Position Error-Aposelene | 255 km | 424 km |
| Velocity Error-Aposelene | 0.003 km/s | 0.005 km/s |

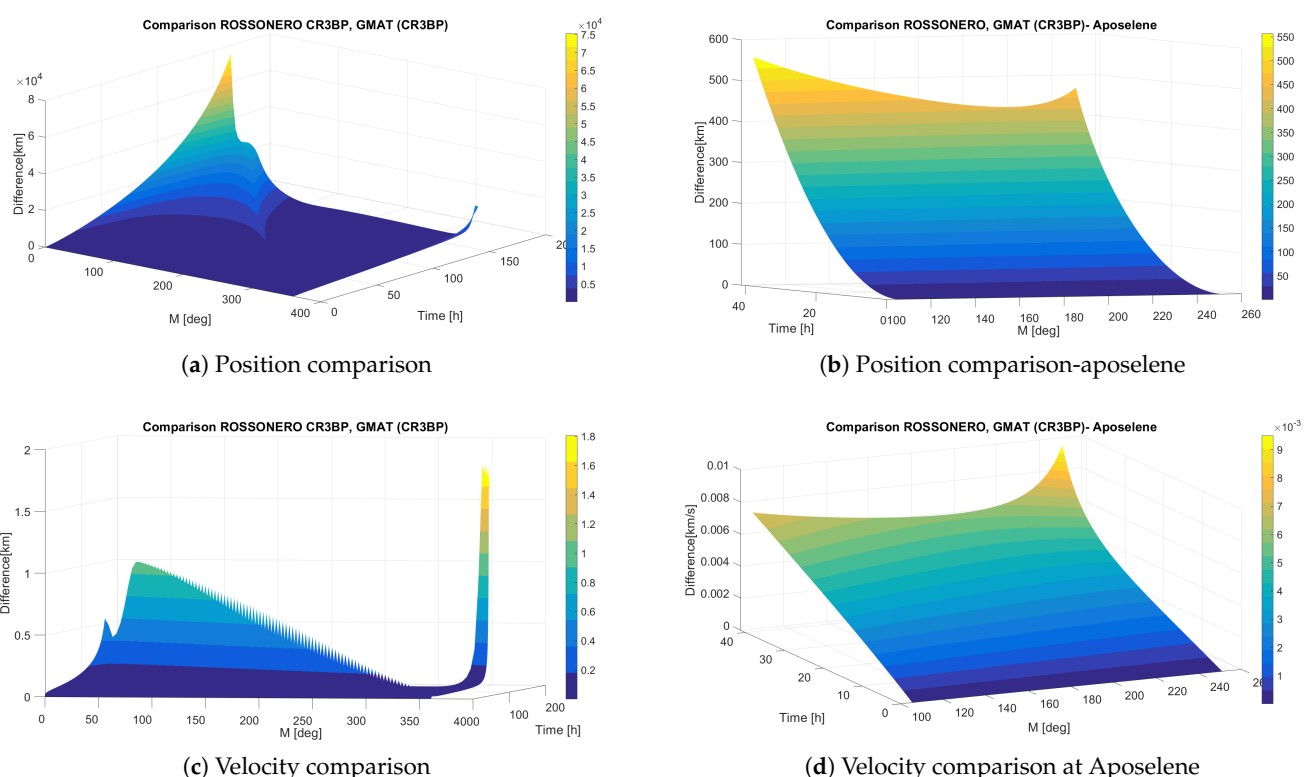

(**a**) Position comparison      (**b**) Position comparison-aposelene

(**c**) Velocity comparison      (**d**) Velocity comparison at Aposelene

**Figure 8.** CR3BP and Ephemeris comparison.

4.1.2. Solar Radiation Pressure (SRP)

Since the presence of the solar radiation pressure SRP was initially neglected in the derivation of the equations of motion, a preliminary evaluation of its influence was also assessed on the dynamics of the single spacecraft and on the relative dynamics. The solar radiation pressure is generated by electromagnetic waves that imprint a pressure on the surfaces hit by the waves themselves. There are many models depending on the spacecraft geometry and orbital properties; see [9], for instance. Here, the pressure is modeled as in [10] and given by:

$$P = \frac{\Phi_E}{c} \tag{19}$$

where $\Phi_E$ is the energy flux associated with the electromagnetic wave spectrum, and $c = 299{,}792{,}458$ m/s is the speed of light. The pressure acts on every exposed surface, and it can influence the motion of the chaser and target vehicles. For additional details, see [10]. In GMAT, the solar radiation pressure model is considered as a spherical radiation from the Sun, with a flux of 1376 W/m². The tests were performed in GMAT with the same Ephemeris model used above, and adding the solar radiation pressure assuming an effective area of 10 m² for the target and 3 m² for the chaser. In the first test, the target free dynamics are propagated and compared with the same dynamics propagated under

the ER3BP; the second test consists of the relative dynamics between chaser and target comparison with the the corresponding dynamics under the ER3BP.

The results are summarized in the figures below. Figure 9 shows the propagation of the target orbit with the influence of the solar pressure compared with the dynamics propagated under the ER3BP. The difference between the two models differs from the ones reported in Figure 7 for the shape but not for the order of magnitude. Therefore, the contribution due to SRP can be considered negligible.

Figure 10 describes the worst case scenario with the orbits propagated in GMAT with and without solar pressure and the same orbit propagated in ER3BP. Figure 11 shows the orbits propagated from the aposelene.

The free motion of the relative dynamics is also compared with the ER3BP with and without the presence of the SRP; see Figures 12 and 13. In the latter, the propagation is performed for one entire orbit, and, as expected, the maximum amount of error is accumulated at the passage at the periselene; after that, the errors decrease again.

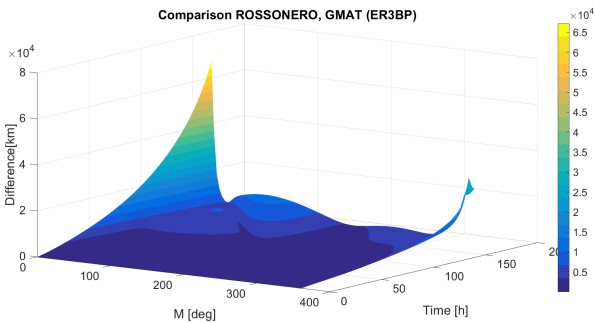

(**a**) Free dynamics comparison with solar radiation pressure

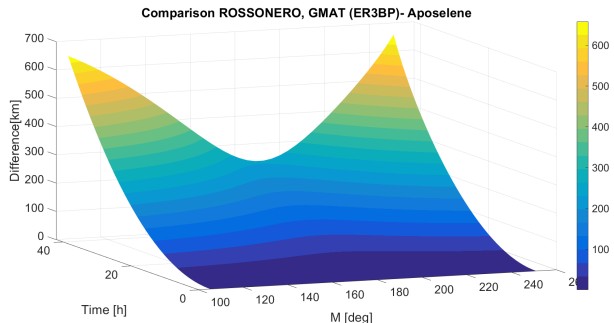

(**b**) Free dynamics comparison with solar radiation pressure, Aposelene

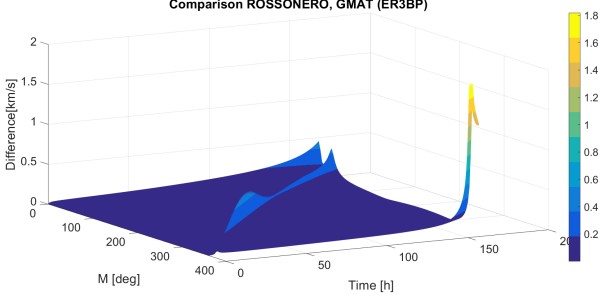

(**c**) Free dynamics comparison with solar radiation pressure, Velocity

**Figure 9.** ER3BP and Ephemeris comparison—Solar radiation pressure.

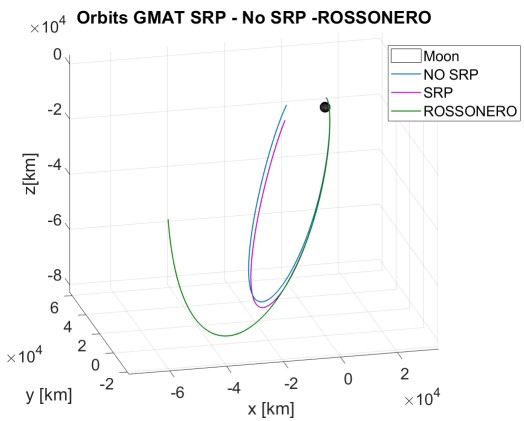

**Figure 10.** Worst case scenario orbit comparison.

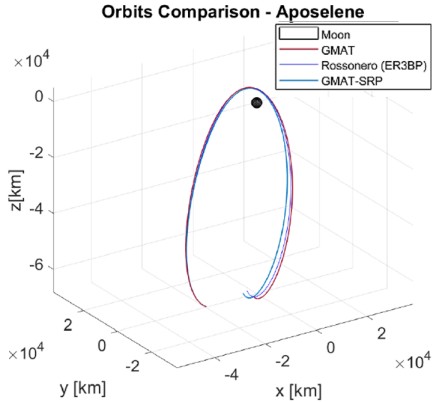

**Figure 11.** Orbit comparison-Aposelene.

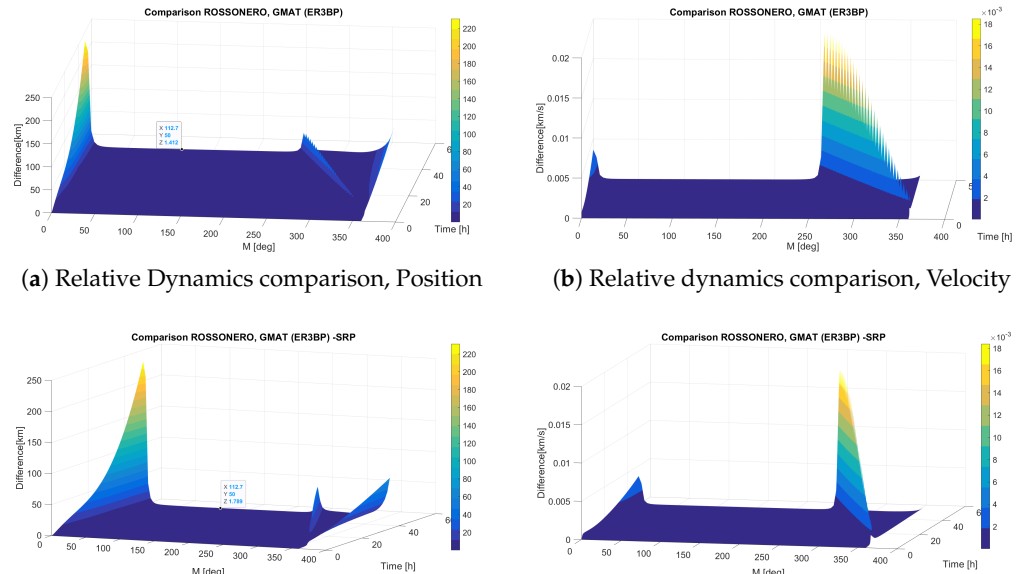

(**a**) Relative Dynamics comparison, Position    (**b**) Relative dynamics comparison, Velocity

(**c**) Relative Dynamics comparison SRP, Position    (**d**) Relative Dynamics Comparison SRP, Velocity

**Figure 12.** Relative dynamics ER3BP- Ephemeris comparison with and without SRP.

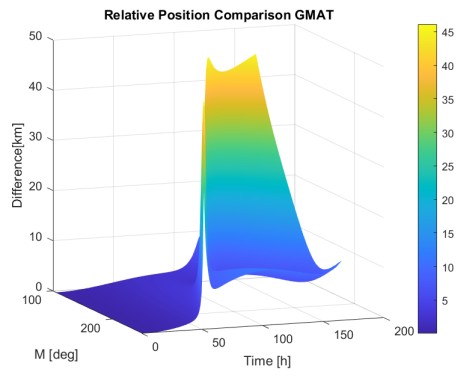

(**a**) Relative Dynamics comparison, Position

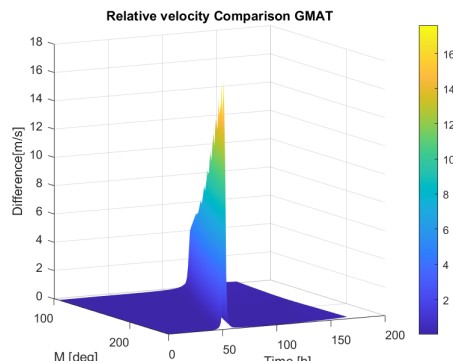

(**b**) Relative dynamics comparison, Velocity

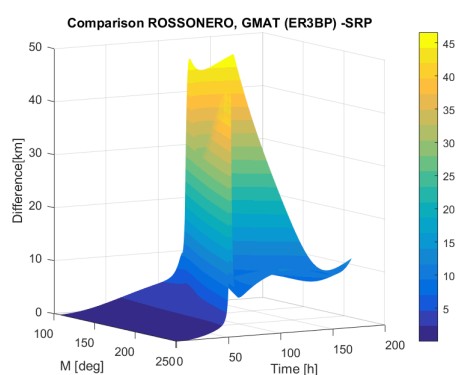

(**c**) Relative Dynamics comparison SRP, Position

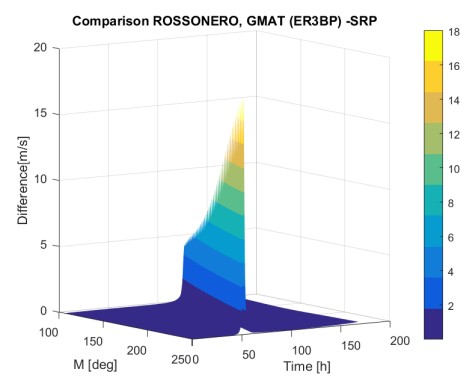

(**d**) Relative Dynamics Comparison SRP, Velocity

**Figure 13.** Relative dynamics ER3BP- Ephemeris comparison with and without SRP.

The presence of the SRP changes the shape of the error of the relative dynamics a bit, but the amplitude of the errors stays in the same order of magnitude. The maximum amount of position error is obtained when the dynamics are propagated starting from the periselene; the maximum amount of velocity error is obtained when the two vehicles pass through the periselene, as expected.

The maximum and minimum relative motion errors with and without SRP are shown in Tables 3 and 4, respectively. The relative errors remain very limited and the maximum position value must be taken into account when designing a trajectory that is safe with respect to imposed keep-out-zones.

**Table 3.** Maximum relative dynamics errors.

|  | **NO-SRP** | **SRP** |
|---|---|---|
| Position Error | 44.95 km | 46.57 km |
| Velocity Error | 17.61 m/s | 18.02 m/s |

**Table 4.** Minimum relative dynamics errors.

|  | **NO-SRP** | **SRP** |
|---|---|---|
| Position Error (1 Orbit) | 8.3 km | 9.09 km |
| Velocity Error (1 Orbit) | $6.2 \times 10^{-5}$ km/s | $5.3 \times 10^{-5}$ km/s |

## 5. Sensors and Actuators' Models

One of the important elements introduced in the equations of motion is a characterization of sensors and actuators. The selection is based on the assumed choice for the rendezvous mission and improves the accuracy of overall dynamic behavior. The following sections provide a general description of component suites, their models, and validity. We remind the reader that the motivation for component selection is based on the relative effectiveness for the mission at hand, rather than a theoretical optimization evaluation. In particular, cameras are selected to play a primary role in the relative position measurement.

### 5.1. Sensors

Due to the characteristics of the mission, the set of sensors used by the guidance and control depends on the relative distance between chaser and target, which selects which suite is active at any particular moment. A qualitative representation is shown in Figure 14.

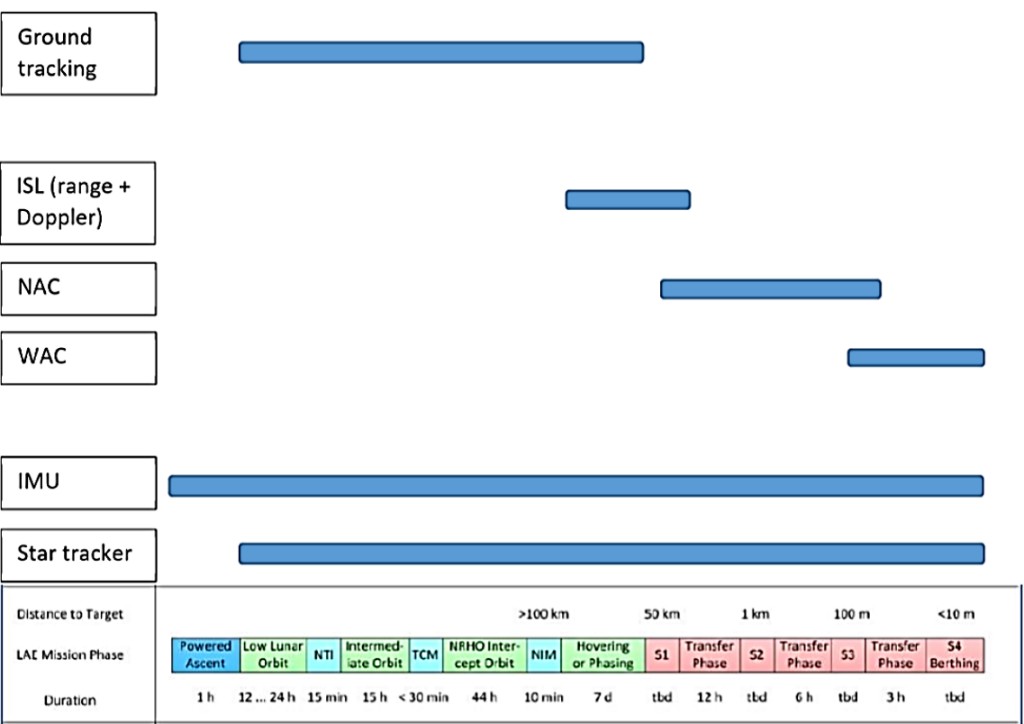

**Figure 14.** Chaser navigation concept.

Of particular interest are:

- Inter Satellite Link
- Wide Angle Camera
- Narrow Angle Camera

ISL, WAC, and NAC were implemented in the model as a single block, with the active sensor selected in relation to the relative distance between chaser and target, with the amount of error committed by the selected sensor's suite used to define the working ranges of each set-up. The main camera parameters are shown in Figure 15.

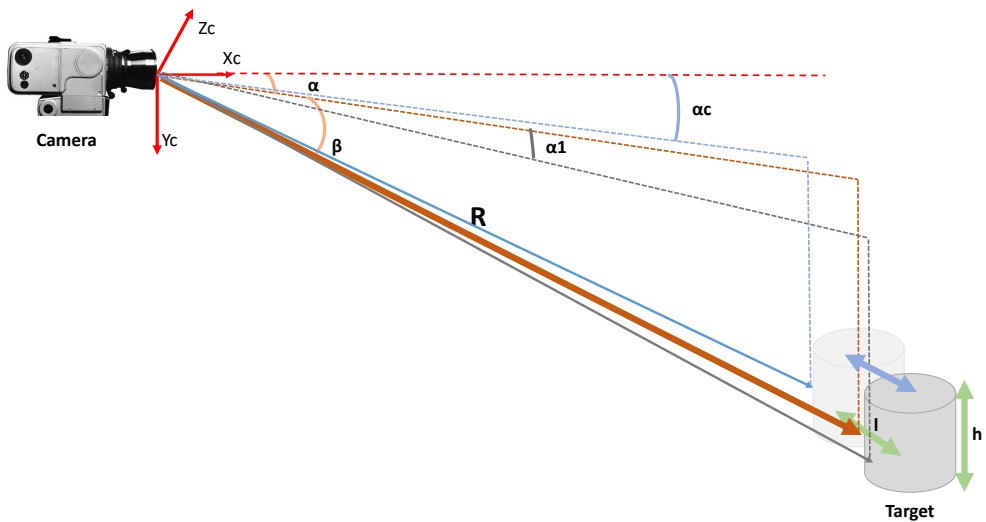

**Figure 15.** Reference quantities and Camera frame.

Range estimation using the NAC is computed according to:

$$RangeError_\% = 0.5\frac{ang_{px}}{2\alpha_1}100 \tag{20}$$

The error is a function of the relative distance $R$ as shown in Figure 16.

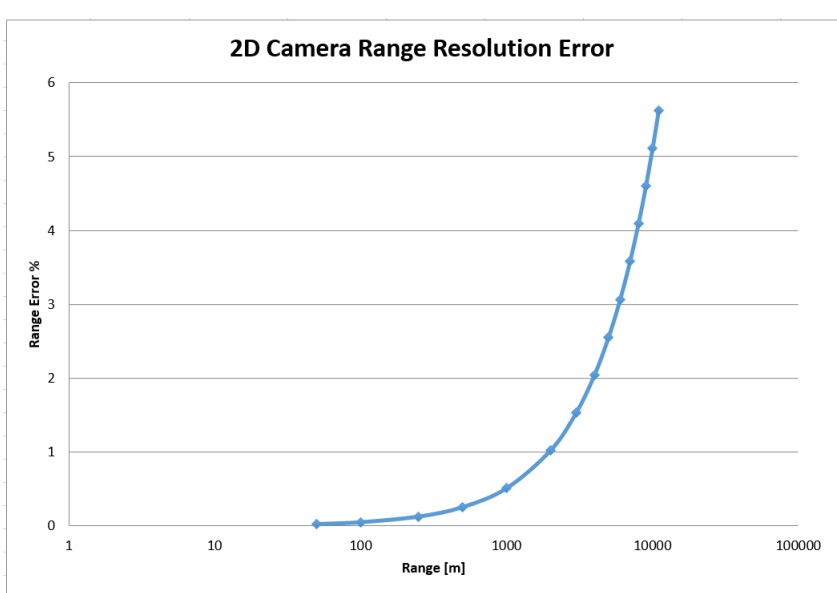

**Figure 16.** Percentage of range error estimate with NAC.

If the distance is above 10 km, the sensor used to estimate the range is the ISL; in fact, it commits an error in the range estimation equal to 3%$R$, and, looking at Figure 16, it is possible to notice that the range estimation error produced by the NAC is above 3% for ranges larger than 6 km. The NAC is also used for the measurement of the lateral displacement ($\alpha$ and $\beta$) and an error of 100% of the target angular visible size is taken into account (about 1–2 px) at large distances.

At medium distances (10 km–1 km), the ISL is deactivated and the range is estimated using the NAC using Equation (21):

$$R_{err} = \frac{l}{2}\left(\frac{\cos \alpha_c}{\tan \alpha_1} + \sin \alpha_c\right) \pm 0.5 \frac{ang_{px}}{2\alpha_1} \tag{21}$$

where:

- $l$ is the target length that is equal to 5 m;
- $\alpha_c$ is the measured azimuth angle of the target center of mass, affected by an error of 100% of the target illuminated size.
- $\alpha_1$ is the difference between the azimuth of the centre of the illuminated zone and the azimuth of a side of the illuminated zone.
- $ang_{px}$ the pixel angular size of the visible Target area.

The lateral displacement is still measured with the NAC and the relative is between 5 and 100 px (100% of the target illuminated side). The lateral displacement error is computed using Equation (22), and the trend is reported in Figure 17.

The angular error $\alpha_{err}$ defines the limits between medium distances and short distances; in fact, medium distances are considered those for which $RangeError_\%$ is less than 3%R, but the lateral displacement error, computed with Equation (22), is less than 100 px; as a result, the medium distances are those included in the range 1–10 km:

$$\alpha_{err} = \arctan \frac{l}{R} \tag{22}$$

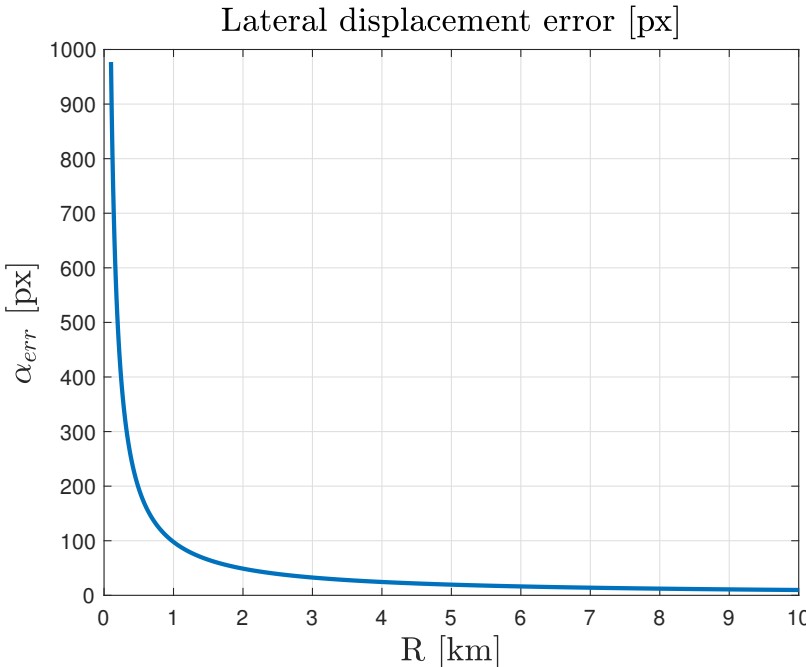

**Figure 17.** Lateral displacement error.

The short distances are considered those between 1 km and 5 m; here, the sensor used is the WAC and the error on lateral displacement is equal to 0.2%R and the error on R is 2%R. At short distances, it is also possible to estimate the relative attitude, with an error of 5 degrees maximum.

The estimation of the translational velocity is based on the differentiation principle and a subsequent Kalman filtering procedure; however, in this work, the entire navigation chain is modeled as in Equation (23):

$$\tilde{\dot{\rho}}_{pp} = \dot{\rho}_{pp} + \|\tilde{\rho}_{pp} - \rho_{pp}\| n_{\dot{\rho}_{pp}} \tag{23}$$

where $\tilde{\dot{\rho}}_{pp}$ is the relative port-to-port velocity affected by the error, $\dot{\rho}_{pp}$ is the velocity without errors, $\tilde{\rho}_{pp}$ is the relative position affected by error, $\rho_{pp}$ is the relative position, and $n_{\dot{\rho}_{pp}}$ is a white noise.

A similar approach was used to model the angular velocity error, as described in Equation (24)

$$\tilde{\omega} = \omega + \|\tilde{\theta} - \theta\| n_\omega \tag{24}$$

where $\tilde{\omega}$ is the relative angular velocity affected by the error, $\omega$ is the velocity without errors, $\tilde{\theta}$ is the relative attitude affected by error, $\theta$ is the relative attitude, and $\mathbf{n}_\omega$ is a white noise. Both $\mathbf{n}_{\dot{\rho}}$ and $\mathbf{n}_\omega$ characteristics can be selected by the user.

The performance of the sensors with the estimated range measurement is shown by the port-to-port trajectory in Figure 18 drawn in red, representative of a V-bar approach to the target.

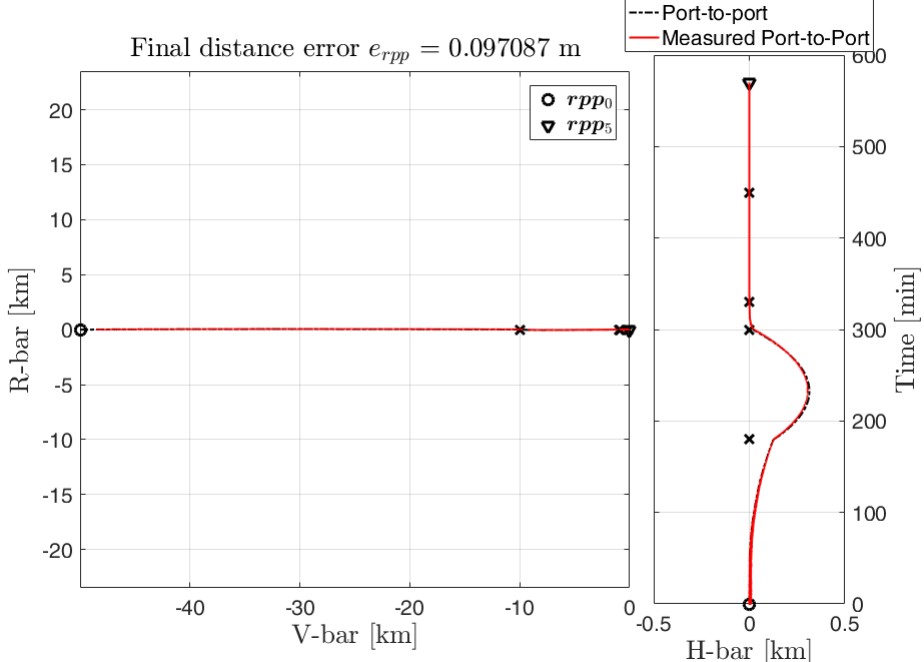

**Figure 18.** Port-to-port maneuver with the measured range.

It is important to remark that the range is measured with the camera only when the the target is within the camera field of view; the sensors are initialized at every hold-point and the sensor suite changes only in the hold-points depending on the target-chaser distance at hold-point. The camera is also used to measure the relative attitude for short distances as shown in Figure 19. The range measurements are always available since we assume the implemented controllers capable of keeping the target within the camera's FOV.

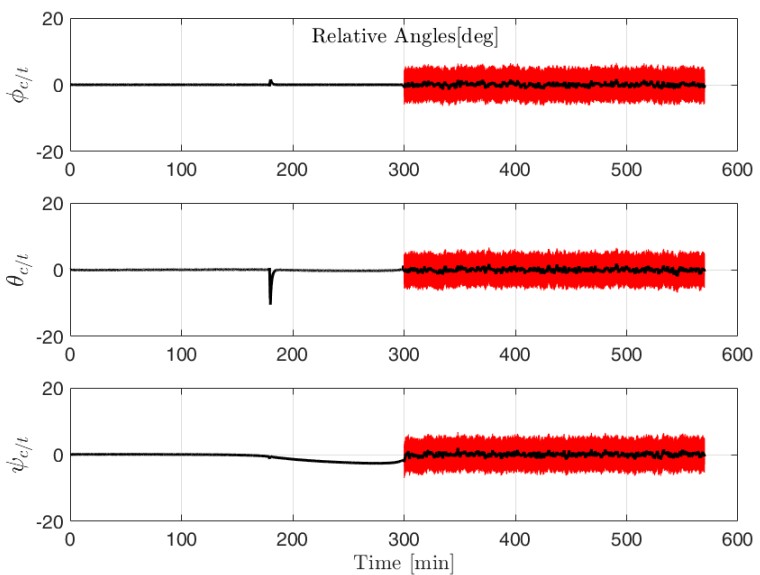

**Figure 19.** Relative angles with their measurements.

*5.2. Actuators*

In general, the set of actuators implements the firing command sequences, and allows the vehicle to separate the motion along different axes and the translational motion from the rotational one. With reference to the Heracles mission, the propulsion is implemented by a main engine and 16 RCS-thrusters of 10N each [11] located on the edges of the chaser vehicle. In this paper, we limit ourselves to describe the model the latter, synthetically shown in Figures 20 and 21. The main sources of errors are thrust magnitude errors, thrust direction errors, rise/fall time, and delays.

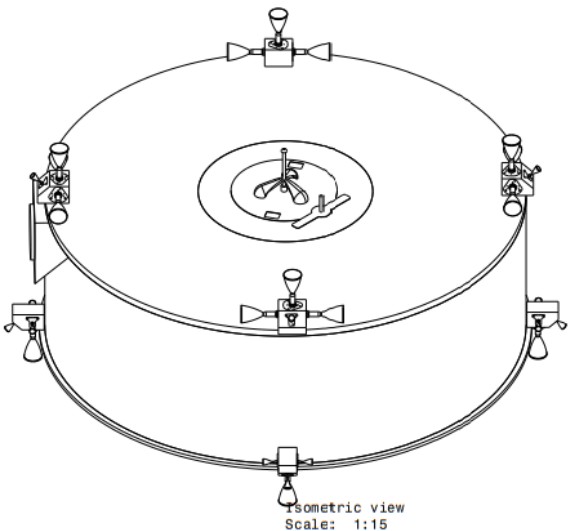

**Figure 20.** 3D CAD rendering of the chaser thruster architecture.

The sixteen engines magnitude can be modeled as:

$$\bar{F} = F \cdot \eta_{1BIT} \cdot (1 + \delta \eta_{1BIT}) \tag{25}$$

where $F$ is the nominal thrust level at steady state condition, $\eta_{1BIT}$ is the theoretical bit efficiency, and $\delta\eta_{1BIT}$ is the impulse bit efficiency random variation. As said before, in our scenario, $F$ is equal to 10N.

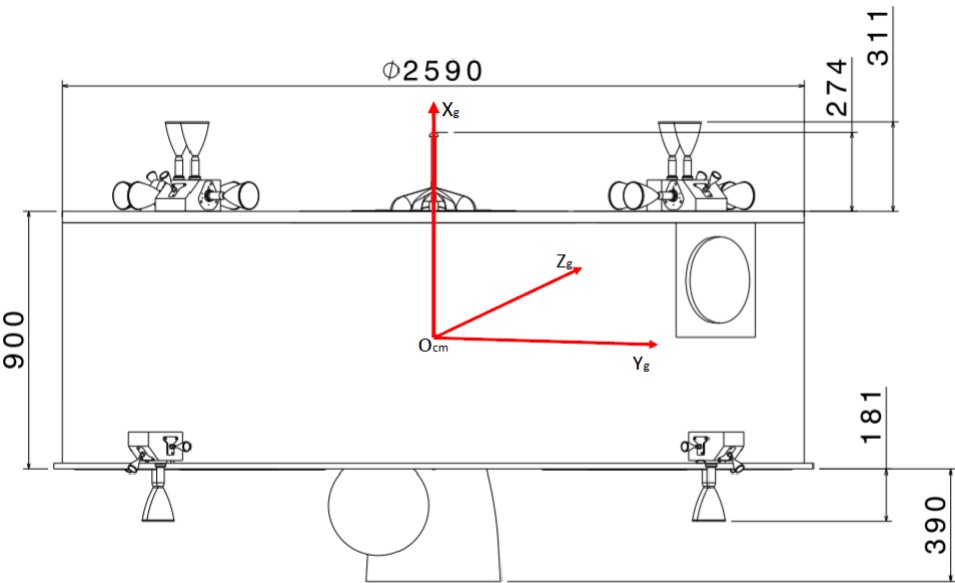

**Figure 21.** CAD rendering of the chaser thruster architecture-side view.

The theoretical impulse bit efficiency is computed from the empirical formula given by Equation (26), which is based on the duration of the thruster firing command $t_{on}$ and the thruster off-time:

$$\eta_{1BIT} = e^{\left(\frac{-c_1}{1000t_{on}+c_2/1000t_{off}}\right)} \tag{26}$$

$t_{on}$ and $t_{off}$ are expressed in seconds, and $c_1$ and $c_2$ are the efficiency factor constants.

According to [12], $t_{on}$ and $t_{off}$ can be selected using a PWM approach. The random variation of the impulse bit efficiency $\delta\eta_{1BIT}$ is assumed to be Gaussian with zero mean and the standard deviation $\sigma_{\eta_{1BIT}}$, given by Equation (27).

$$\sigma_\eta = min(max(\sigma_{\eta_{min}}, \sigma_{\eta_{coef1}} \cdot (1000t_{on})^{\sigma_{\eta_{coef2}}}), \sigma_{\eta_{sat}}) \tag{27}$$

The misalignment is mainly composed of two contributions: internal and external.

The internal error is due to a misalignment between thrust vector and flange. They are constant over a single thruster firing, but they change randomly from one firing to the next, so the internal uncertainty is assumed to behave as a Gaussian noise. The external errors are due to some misalignment between flange and master reference cube.

The uncertainties remain constant for the entire simulation, but they change from a simulation to the other. As a consequence, the external uncertainty is assumed to behave as a uniformly distributed variable.

In addition to magnitude and direction uncertainty, it is necessary to also take into account the Rise/Fall time behavior [7]. The thrust is discretized following a PWM logic. The motors have a dynamic and minimum $t_{on}$ and $t_{off}$ times. The Rise/fall times are taken into account through a filtering operation: the PWM signal will be filter by a 1st or 2nd order filter according to Equations (28) and (29). In the filter equation, we can also take into account a time delay $\tau$:

$$F_1 = \frac{A}{s+b}e^{-\tau} \tag{28}$$

$$F_2 = \frac{\omega_n^2 e^{-sT_{\Delta v}}}{s^2 + 2\zeta\omega_n s + \omega_n^2}e^{-\tau} \tag{29}$$

The location of the thrusters depends on the vehicle, its configuration, geometry, and mission. Thus, the contribution that each actuator gives to torques and forces can be computed with respect to the chaser center of mass in the body frame axis, which is the *Geometrical frame*, defined earlier. Due to the absence of data, the chaser vehicle is considered as a perfect cylinder with uniform density and constant mass; therefore, the body reference system—that coincides with $\mathcal{G}$ in this case—was located along principal axes of inertia.

The relationship between the thrust of each motor and the torques and forces produced is then computed by projection. Let us define: $\mathbf{F}_s$, the vector that contains all the thrust provided by the small thrusters:

$$\mathbf{F}_s = \begin{bmatrix} F_1 & F_2 & ... & F_{16} \end{bmatrix}^T \tag{30}$$

The 6x1 vector of forces and torques, in chaser body frame, is called $\tau$, and it is defined as:

$$\tau = \begin{bmatrix} F_x & F_y & F_z & N_x & N_y & N_x \end{bmatrix}^T \tag{31}$$

The relation between $\tau$, $F_s$ and $F_b$ is a linear relationship given by Equation (32):

$$\tau = [B_s][F_s] \tag{32}$$

where $B_s$ is called the $6 \times 16$ control allocation matrix [5].

Control allocation is an area with a large amount of literature and many algorithmic techniques, spanning from structural geometry approaches, to several types of optimization methods. Ref. [13] is especially interesting for our application; in fact, it avoids the use of a PWM module after the control allocation because it computes the optimal combination of duty-cycle to minimize a linear cost function given by:

$$J = f^T[\delta_1, \ldots, \delta_n], \; f = [1 \; 1 \ldots 1]^T \tag{33}$$

where $[\delta_1, \ldots, \delta_n]$ is the vector that contains the duty-cycles; each element of this vector is included between 0 and 1, and $F_s = F_m ax[\delta_1, \ldots, \delta_n]$, where $F_m ax$ is the maximum thrust supplied by a thruster. It defines the control law in (32).

Another advantage of this technique is that, by minimizing the duty-cycle vector, the fuel-consumption is minimized as well because it is proportional to the summation of all thruster on time. For the reasons briefly explained above, also supported by empirical results, one of the selected control allocation techniques for comparison is the one proposed by [13]. Clearly, the selection of a specific allocation algorithm also depends on safety factors and software computational power specific for space applications. Therefore, a look-up table approach was also investigated in this study.

The control allocation methods described above are described in detail in Refs. [13,14], and the interested reader is referred to them. In the following, we only present rendezvous simulation results as a function of PWM working frequency (1 Hz and 2 Hz, respectively). The optimal control allocation in Equation (33) selects an optimal combination of Pulse Width Modulation duty cycle at each step—in the sense that it minimizes the summation over all duty-cycles and, consequently, the fuel consumption. The optimization method is the interior-point-legacy (default) and the optimization procedure was executed at each step using available routines. The rendezvous maneuver simulation is shown in Figure 22, indicating better performance with a higher PWM frequency.

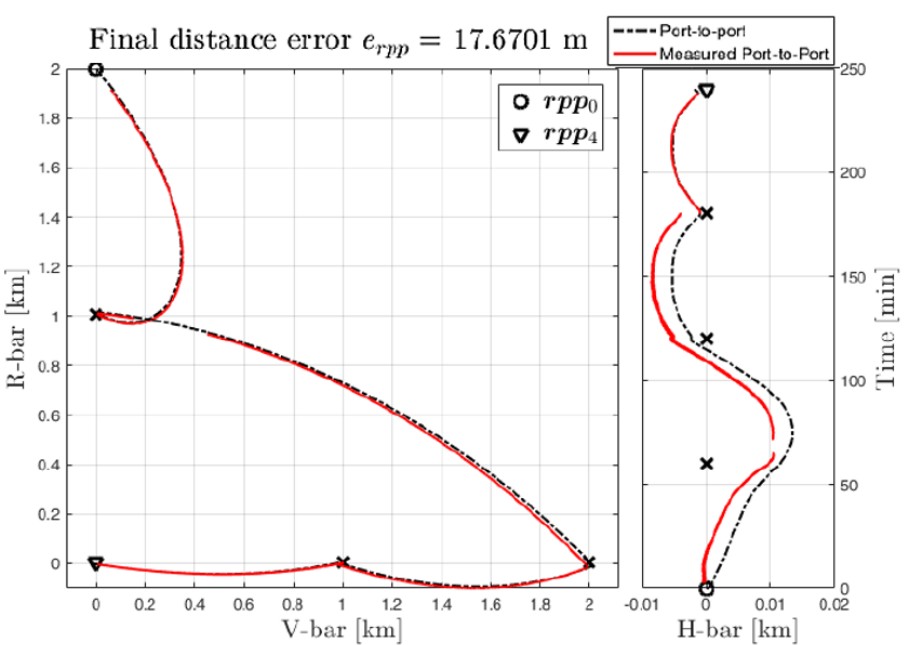

**(a)** Port-to-port trajectory, $T = 1s$

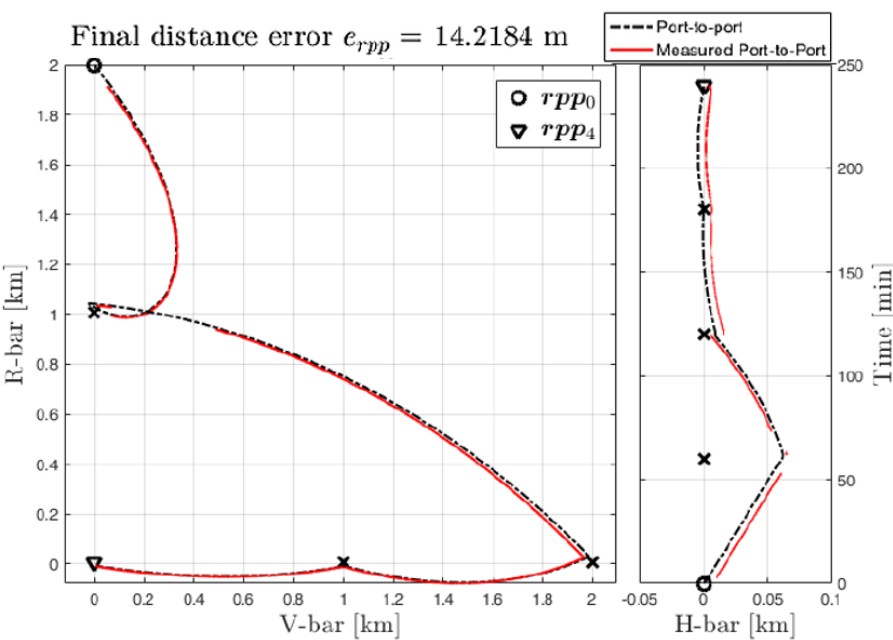

**(b)** Port-to-port trajectory, $T = 0.5s$

**Figure 22.** Trajectories varying the PWM frequency (optimal allocation).

In the look-up table algorithm, the thruster modulator is the on-board function that computes parameters for the actuation of the thrusters during a control cycle. The main objective is to compute the percentage of the total duration of the control cycle for which each thruster must be open such that that the average effect of the thrusters over the control cycle results in a force and torque as requested by the controller. The resulting trajectories are shown in Figure 23, for the same frequency range.

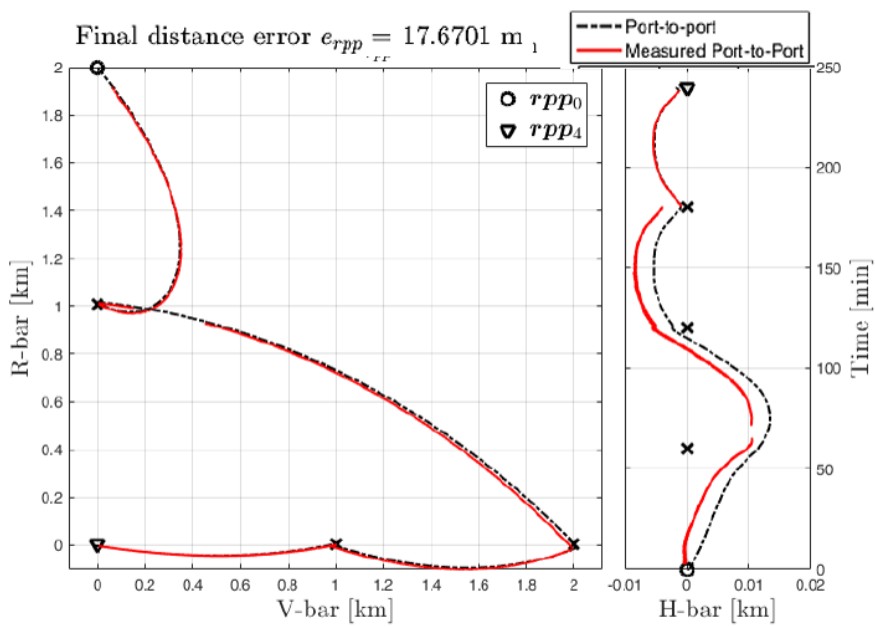

**(a)** Port-to-port trajectory, $T = 1s$

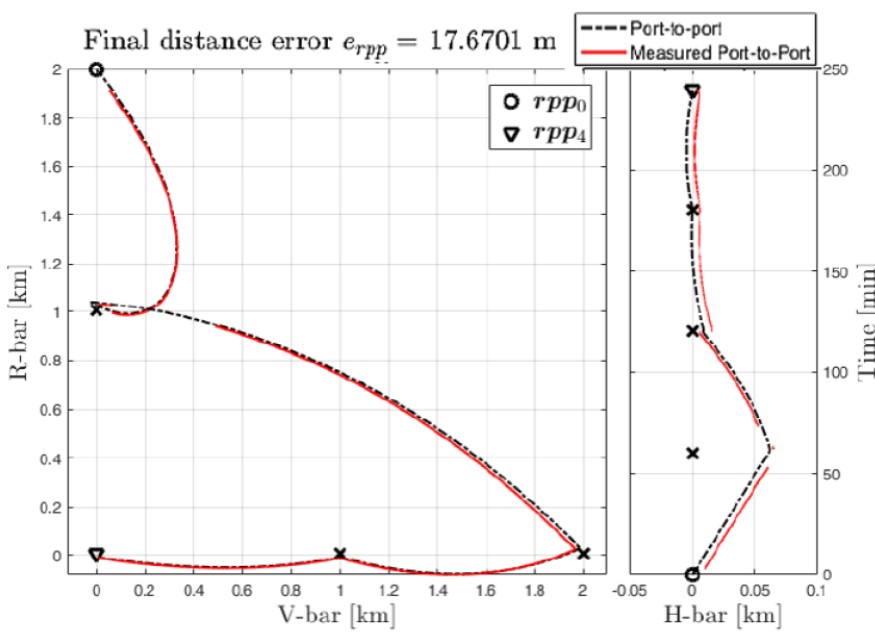

**(b)** Port-to-port trajectory, $T = 0.5s$

**Figure 23.** Trajectories varying the PWM frequency (Look-Up table).

Using the Heracles mission data, a comparison in terms of thrust consumption is shown in Table 5. The table displays the integral of the acceleration at the thruster during the entire maneuver duration. The "ideal" implies a simulation performed using perfect sensors and actuators, while "non-ideal" indicates results obtained with more accurate models for sensors and actuators as described earlier.

**Table 5.** Thrust comparison.

| Thrust Consumption | |
| --- | --- |
| Optimal allocation ideal | 15.4 m/s |
| Optimal allocation non-ideal | 16.1 m/s |
| LUT ideal | 22.1 m/s |
| LUT non-ideal | 27.1m/s |

## 6. Rendezvous Guidance and Control

A rendezvous guidance profile is generally composed of several elements aimed at steering the chaser towards the target, moving between a set of control points. In the Keplerian case, these trajectories are the result of the application of impulsive or continuous thrust firings along prescribed directions. Except for very small relative distances, the maneuvers are computed in an open-loop fashion, and can be based on the solution of the Hill's equations [4], for instance. The accuracy of the Hill's equations for the restricted-three-body problem is poor and may compromise the good execution of the maneuver, especially due to their open-loop implementation; therefore, in this section, some of the maneuvers used for rendezvous and docking are reproduced exploiting the linearized elliptic ELERM and circular CLERM equations, whose derivation can be found in [2], together with their range of validity along the target orbit.

The linearization procedure yields a linear time varying expression in state variable form given by:

$$\dot{x} = A(t)x + Bu \tag{34}$$

with $A \in \mathbb{R}^{6\times6}$ defined as:

$$A(t) = \begin{bmatrix} \mathbf{0}_{3\times3} & I_3 \\ A_{\dot{\rho}\rho}(t) & -2\Omega_{l/i}(t) \end{bmatrix} \tag{35}$$

$$A_{\dot{\rho}\rho} = -\left[\dot{\Omega}_{l/i}\right]_{\mathcal{L}} - \Omega_{l/i}^2 - \frac{\mu}{r^3}\left(I - 3\frac{rr^T}{r^2}\right) - \frac{1-\mu}{\|r+r_{em}\|^3}\left(I - 3\frac{(r+r_{em})(r+r_{em})^T}{\|r+r_{em}\|^2}\right) \tag{36}$$

with state variable defined in Equation (7). The difference between ELERM and CLERM in Equation (34) is due to the angular velocities assumption between elliptic and circular motions.

The closed-form solution of the these equation sets is not straightforward, due to the presence of time-varying parameters, and the absence of general analytical solutions for ER3BP and CR3BP orbits. To deal with this problem, different guidance approaches are introduced. The selection of the guidance algorithm is influenced by the relative distance, the required level of accuracy, the capability of closing the loop, and the available/current sensor's and actuator's suite. We must remark that the aim of the paper is not to propose the best guidance algorithm, rather to provide a suitable choice for the mission. Figure 24 shows a sequence of algorithms that are used in the paper. The mathematical details are reported in the Appendix A. The methods used for the guidance are:

- Adjoint Guidance.
- PID.
- SDRE.

The first two techniques are used to control the relative translational motion and the absolute attitude motion outside of a safety area defined to avoid collision, and they operate in an open loop fashion for large to medium distances. The third controller takes care of the regulation of the attitude and the position in the last part of the approach.

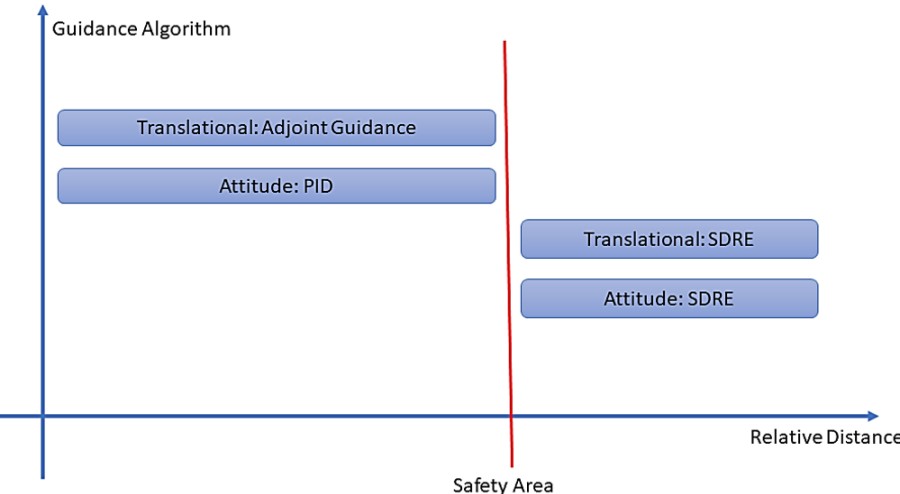

**Figure 24.** Schematic explanation of the guidance sequence.

### 6.1. Adjoint Guidance

Adjoint models (or more properly *dual models*) have been extensively exploited for the solution of classical missile guidance problems, see [15,16], and more recently in [17,18]. They appear to be well suited due to the time varying nature of the relative dynamics. The application of adjoint theory to our problem is described in Appendix A.1.

The main idea is to consider the relative dynamics as described by ELERM (or in a simpler model CLERM) as given by Equation (A1), with time response at some finite time $t_f > t_0$ given by Equation (A3). We can associate with Equation (A1) a dual system at time $t_f$ [19] given by Equation (A4), whose initial time response coincides with the final time response of the original system.

The adjoint guidance is applied in this work to a two-impulse maneuver at each hold point. In other words, the chaser leaves a hold point with an impulsive firing and reaches the next with a braking impulse. The total input has the form given by Equation (A8) with two impulses applied at times $t_1$ and $t_2$; then, we obtain relative position and rate from Equation (A15), where $(\rho_f, \dot{\rho}_f)$ is the desired relative state, computed starting from ELERM.

The performance of the guidance was evaluated computing relative distance and velocity errors repeated here for clarity's sake:

$$e_{\rho} = \|\rho_f - \rho(t_f)\|, \quad e_{\dot{\rho}} = \|\dot{\rho}_f - \dot{\rho}(t_f)\| \tag{37}$$

with total control effort:

$$\Delta V = \int_{t_0}^{t_f} \|u(t)\| \mathrm{d}t \tag{38}$$

### 6.2. PID Guidance Controller

During the initial phase of the rendezvous, the primary concern is to follow a prescribed attitude profile, or to maintain the target in the field of view of the chaser's cameras. To this end, a standard PID controller on all axes was considered to be satisfactory. The controllers were simply tuned to obtain a good rise time.

An example of the performance without and with the controller is shown in Figure A1 and Figure A2, respectively. The simulation considers a nonzero initial condition on the angular velocity with no damping at the beginning of the maneuver. The continuous rotation of the vehicle can be seen in Figure A1. After the insertion of the controller, the vehicle is stabilized to the desired attitude (zero in this case) as shown in Figure A2. The presence of the PID also allows the pointing of the cameras always towards the target

avoiding possible singularity issues (of course, the equations can be always transformed in their quaternion, not changing the general controller philosophy).

Two different attitude-modes are available for large and medium distances, where the PID is in charge of controlling the attitude of the chaser. In the former case, the chaser is forced to follow a user-defined attitude profile w.r.t the LVLH; this profile is defined in Euler angles. In the latter case, the chaser must point always towards the target to keep it inside the camera FOV, with the dynamics expressed either in Euler angles or quaternions [20].

In the latter, the quaternion ($q_{ref}$) that represents the angular displacement between the camera axis and the target position w.r.t. the chaser being computed first. Then, it is used as a reference for the controller; this means that the controller is asked to zero the chaser attitude w.r.t $q_{ref}$. To compute $q_{ref}$, let us define $\mathbf{n}_{ref}$ and $\theta_{ref}$ as in Equations (39) and (40):

$$\mathbf{n_{ref}} = \frac{\mathbf{x_c} \times (-\boldsymbol{\rho})}{\|\mathbf{x_c} \times (-\boldsymbol{\rho})\|} \tag{39}$$

$$\theta_{ref} = \arctan \frac{\|\mathbf{x_c} \times (-\boldsymbol{\rho})\|}{\mathbf{x_c} \cdot (-\boldsymbol{\rho})} \tag{40}$$

Combining the equations yields:

$$q_{ref} = [\cos(\frac{\theta_{ref}}{2}); \mathbf{n}_{ref} \sin(\frac{\theta_{ref}}{2})] \tag{41}$$

The quantities written in Equations (39)–(41) are graphically represented in Figure 25.

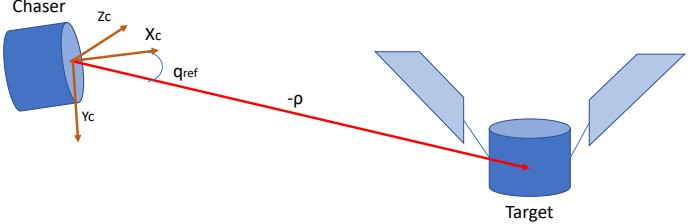

**Figure 25.** Target pointing strategy.

At close distances, the user can only define the desired relative attitude profile; in other words, the user can define the attitude of the chaser w.r.t. the target at each hold-point that is located at a close distance from the target.

*6.3. SDRE Guidance*

Since the middle of the 1990s, State-Dependent Riccati Equation (SDRE) strategies have emerged as a general design structure that provide a systematic and effective means of designing nonlinear controllers, observers, and filters. These methods overcome many of the difficulties and shortcomings of existing methodologies, and deliver computationally simple algorithms that have been highly effective in a variety of practical and meaningful applications. In a special session at the 17th IFAC Symposium on Automatic Control in Aerospace 2007, theoreticians and practitioners in this area of research were brought together to discuss and present SDRE-based design methodologies as well as review the supporting theory. It became evident that the number of successful simulations, experimental, and practical real-world applications of SDRE control have outpaced the available theoretical results.

The methodology originated as an extension of Linear Quadratic Control techniques to a special class of nonlinear systems. An interesting survey on the method can be found in [21]. Stability and controllability aspects are discussed in [22,23]. SDRE controllers were

used in space related applications, especially for the control of proximity operation and formation flight as reported in [24–26].

The method entails a factorization (that is, parameterization) of the nonlinear dynamics into a state vector and the product of a matrix-valued function that depends on the state itself. In doing so, the SDRE algorithm fully captures the nonlinearities of the system, bringing the nonlinear system to a (non unique) linear structure having state-dependent coefficient matrices, and minimizing a nonlinear performance index having a quadratic-like structure. A state dependent algebraic Riccati equation P using the SDC matrices is then solved online to give the suboptimum control law.

A more detailed derivation can be found in Appendix A.3. The structure of the control law, in its simpler form that assumes the measurement of the entire state, is a full state feedback controller given by Equation (A25):

$$\mathbf{u} = -\mathbf{R}^{-1}(\mathbf{x})\mathbf{B}^\top(\mathbf{x})\mathbf{P}(\mathbf{x})\mathbf{x}$$

The procedure can be applied in the presence of inequality constraints as well, derived from a set of admissible states on relative position and velocity, especially at close distances, where relative motion precision is necessary as shown in Equation (A26). The resulting control law then becomes:

$$\mathbf{u} = -\mathbf{K}(\mathbf{x})\mathbf{x}$$
$$= -\big[\mathbf{K}_0(\mathbf{x}) + \mathbf{K}_\Omega(\mathbf{x})\big]\mathbf{x}$$

Here, the gain matrix includes the constraint term $K_\Omega$. A constrained proximity representation is shown in Figure A3 of Appendix A.3, with the constraints derivation.

A numerical simulation is shown below using the initial conditions in Table 6. We assume that the attitude motion of the target has a maximum amplitude of 5°. For the chaser, the inertia matrix is $\mathbb{I} = 10^{-3} \cdot diag(1.1, 0.6, 0.6)$ kg · m². The direction vector of approaching cone is $[\mathbf{p}]_\mathcal{T} = [-1, 0, 0]^\top$, and the maximum cone angle is set as $\beta = 25°$.

**Table 6.** Initial attitude condition.

| Variable | Data |
| --- | --- |
| $\mathbf{q}_{c/l}$ | $[1.0, 0, 0, 0.0087]^\top$ |
| $\boldsymbol{\omega}_{c/i} \left[\mathrm{rad\,s^{-1}}\right]$ | $[0, 0.01, 0]^\top$ |
| $\mathbf{q}_{t/l}$ | $[0.9999, -0.0061, -0.0061, -0.0061]^\top$ |
| $\boldsymbol{\omega}_{t/l} \left[\mathrm{rad\,s^{-1}}\right]$ | $[0.0019, 0.0019, 0.0019]^\top$ |
| $[\boldsymbol{\rho}_0]_\mathcal{L}$ [km] | $[-10, 0, -4]^\top$ |
| $[\dot{\boldsymbol{\rho}}_0]_\mathcal{L} \left[\mathrm{km\,s^{-1}}\right]$ | $[0, 0, 0]^\top$ |

The example was selected to show the performance of the SDRE guidance, assuming perfect sensor and actuator models. The resulting trajectory is shown in Figure 26.

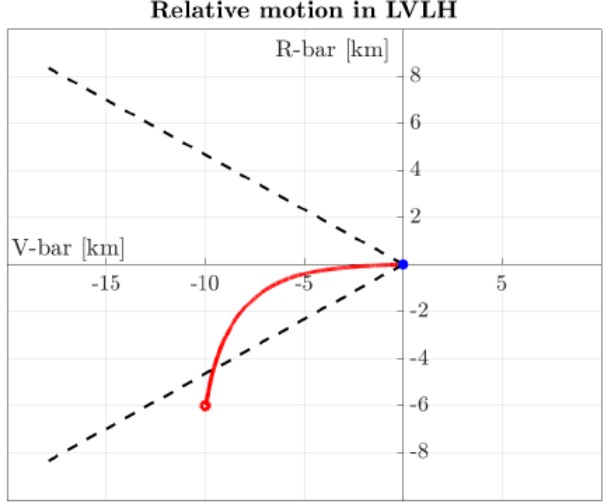

**Figure 26.** Approaching constraint example.

## 7. Full Rendezvous Sequence

This section describes an example of full rendezvous/berthing maneuver, using the methodologies described in the paper.

As shown in Section 4, the simulation uses the Ephemeris model, in particular the DE430 provided by JPL. The hold-point settings for the maneuver are given by:

- HP0 = [−50; −7.6024; 5.9946] km, TOF 40 h;
- HP1 = [−11.4573; −7.6024; 5.9946], TOF 12 h;
- HP2 = [−0.9003; −1.9097; 0.4777], TOF 8 h;
- HP3 = [−0.1; 0; 0], TOF 6 h;
- HP4 = [0; 0; 0];

The location of the hold-points is passively safe [5], and it was selected minimizing the energy consumption. The duration of the different transfer times is designed to enter the safety area at the aposelene; moreover, the location of HP0 within the target orbit corresponds to a mean anomaly of about 113° that guarantees one of the lowest differences between the Ephemeris equations and the ER3BP equations (inclusive of sensor and actuator dynamics).

Figure 27 shows the 3D full rendezvous trajectory, Figure 28 show its the R-bar–V-bar projection, with the safety zone indicated by the circle satisfied until actual berthing occurs.

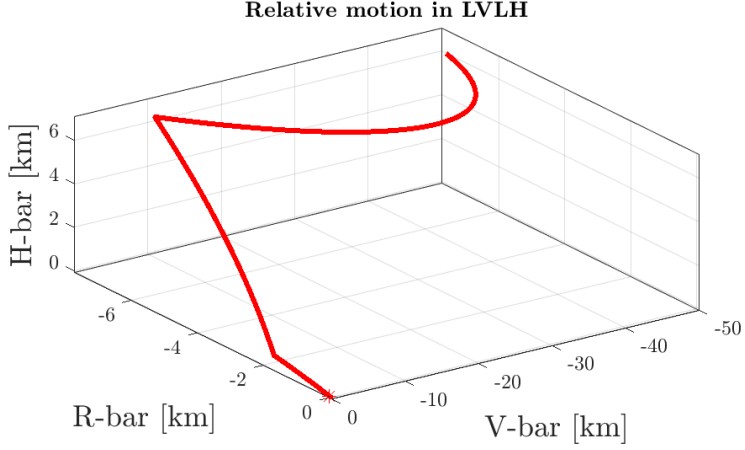

**Figure 27.** Rendezvous trajectory.

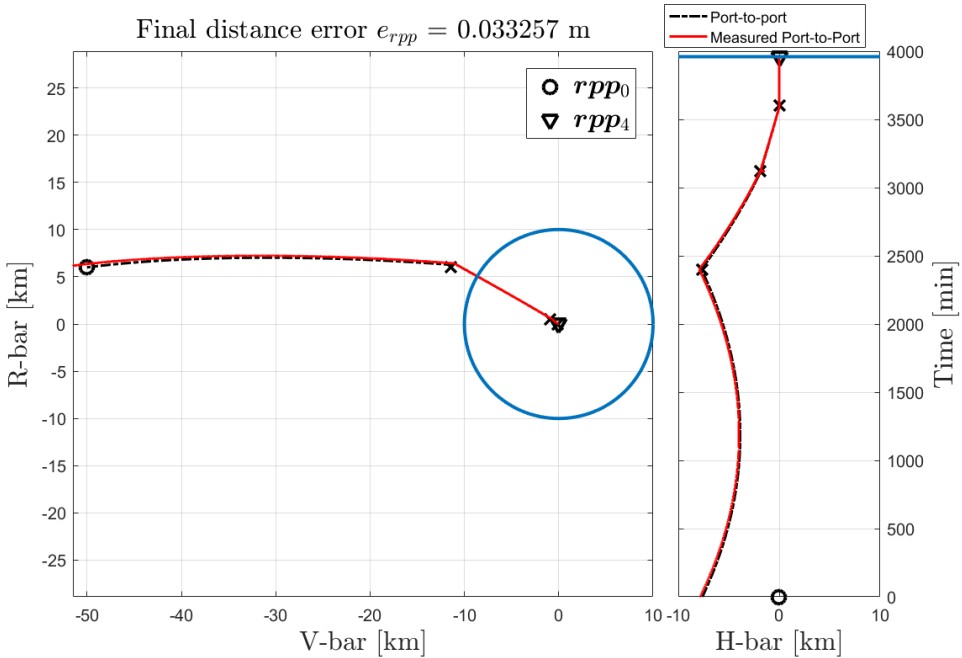

**Figure 28.** Rendezvous trajectory-LVLH.

Figure 29 shows the relative velocity components between chaser and target. The relative attitude is shown in Figure 30. In both cases, the contribution of sensors and actuators models is shown in red as compared to the ideal case (black line).

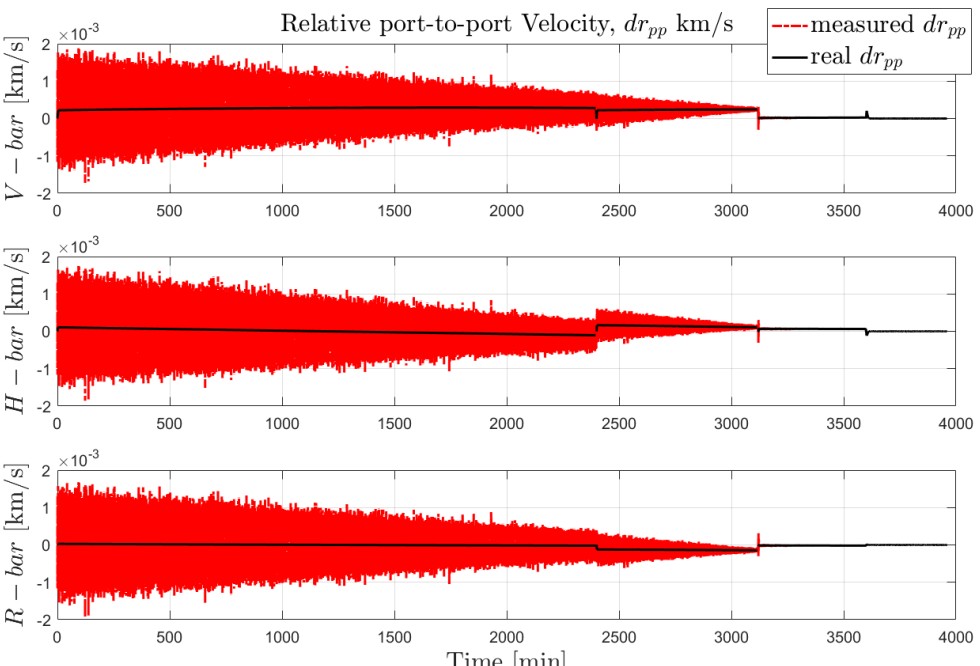

**Figure 29.** Relative rendezvous velocity.

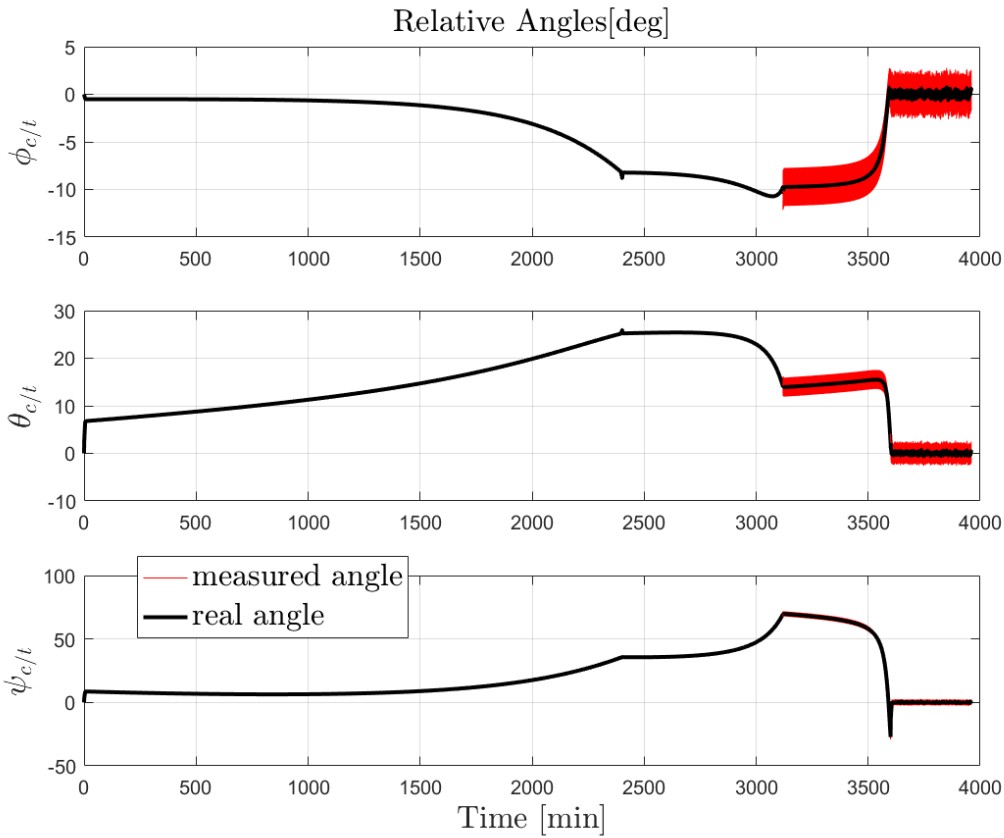

**Figure 30.** Relative attitude.

## 8. Conclusions

The paper presents a comprehensive preliminary analysis of the dynamics and control issues arising in a rendezvous mission between an active vehicle departing from lunar orbit to a target station located in an NRHO L2 orbit. The mission requires a three-body problem description due to the gravity of both Earth and Moon. The implementation of dynamic components was based on ESA's Heracles mission data and preliminary configuration. The accuracy of the model was improved with the inclusion of sensors and actuators models, and guidance algorithms were selected in order to verify the reliability of the GNC loop. The validation was presented using an Ephemeris model. The paper shows that elliptic restricted three-body dynamics offers a good approximation; in fact, since the rendezvous will occur at the aposelene, the circular restricted model could be used as a starting point for the guidance design. If the first rendezvous were aborted, the mall errors in the relative motion would allow a try after one orbit. The selection of guidance algorithms was driven by feasibility, and a full mission profile was tested successfully in simulation.

**Author Contributions:** The authors contributed equally. All authors have read and agreed to the published version of the manuscript.

**Funding:** This work was partially supported by the European Space Agency under contract No. 000121575/17/NL/hh. The view expressed herein can in no way be taken to reflect the official opinion of the European Space Agency.

**Institutional Review Board Statement:** Not applicable.

**Informed Consent Statement:** Not applicable.

**Data Availability Statement:** Not applicable.

**Conflicts of Interest:** The authors declare no conflict of interest.

## Abbreviations

| | |
|---|---|
| LAE | Lunar Ascent Element |
| LOP-G | Lunar Orbital Platform-Gateway |
| NRHO | Near Rectilinear Halo Orbit |
| ISS | International Space Station |
| TOF | Time of Flight |
| GNC | Guidance Navigation and Control |
| LVLH | Local Vertical Local Horizon |
| ER3BP | Elliptic Restricted Three Body Problem |
| CR3BP | Circular Restricted Three Body Problem |
| CoM | Center of Mass |
| GMAT | General Mission Analysis Tool |
| SRP | Solar Radiation Pressure |
| ISL | Inter Satellite Link |
| WAC | Wide Angle Camera |
| NAC | Narrow Angle Camera |
| FOV | Field of View |
| RCS | Reaction Control System |
| ELERM | Elliptic Linearized Equations of Motion |
| CLERM | Circular Linearized Equations of Motion |
| SDRE | State Dependent Riccati Equations |
| SDC | State Dependent Coefficient |

## Appendix A. Guidance Algorithms Derivation

The appendix describes some of the mathematical derivations relative to the algorithms and methods used for devising the guidance loops used in the paper.

### Appendix A.1. Adjoint Method Theory

Adjoint models (or more properly *dual models*) have been extensively exploited for the solution of classical guidance problems, see, e.g., [15,16], and more recent advances in [17,18]. They appear to be well suited due to the time varying nature of the relative dynamics.

Consider a linear time-varying system $\boldsymbol{\Sigma}$, with state $\boldsymbol{x} \in \mathbb{R}^n$, output $\boldsymbol{y} \in \mathbb{R}^h$, and control vector $\boldsymbol{u} \in \mathbb{R}^k$:

$$\boldsymbol{\Sigma}: \begin{cases} \dot{\boldsymbol{x}}(t) = \boldsymbol{A}(t)\boldsymbol{x}(t) + \boldsymbol{B}(t)\boldsymbol{u}(t) & \text{(A1)} \\ \boldsymbol{y}(t) = \boldsymbol{C}(t)\boldsymbol{x}(t) & \text{(A2)} \end{cases}$$

with initial condition $\boldsymbol{x}(t_0) = \boldsymbol{x}_0$. The matrices $\boldsymbol{A}(t) \in \mathbb{R}^{n \times n}$, $\boldsymbol{B}(t) \in \mathbb{R}^{n \times k}$, $\boldsymbol{C}(t) \in \mathbb{R}^{h \times k}$ are assumed to be matrix-valued smooth functions. The system response at time $t_f > t_0$ is given by:

$$\boldsymbol{y}(t_f) = \boldsymbol{C}(t_f)\boldsymbol{\Phi}(t_f, t_0)\boldsymbol{x}_0 + \int_{t_0}^{t_f} \boldsymbol{K}(t_f, \tau)\boldsymbol{u}(\tau)\mathrm{d}\tau \tag{A3}$$

where we introduced the *state transition matrix* of the system $\boldsymbol{\Phi} \in \mathbb{R}^{n \times n}$, and the *impulse response matrix* $\boldsymbol{K} \in \mathbb{R}^{h \times k}$ that represents the response of the system to the impulse, for $\boldsymbol{x}_0 = \boldsymbol{0}$. More specifically, the element $k_{ij}(t, \tau) = \{\boldsymbol{K}(t, \tau)\}_{ij}$ is the response at time $t$ on the output $y_i$ for an impulse applied on $u_j$ at time $\tau > t_0$.

We now introduce the *dual* of system $\boldsymbol{\Sigma}$ at time $t_f$ [19]:

$$\overline{\boldsymbol{\Sigma}}: \begin{cases} \dot{\overline{\boldsymbol{x}}}(\eta) = \boldsymbol{A}(t_f - \eta)^T \overline{\boldsymbol{x}}(\eta) + \boldsymbol{C}(t_f - \eta)^T \overline{\boldsymbol{u}}(\eta) & \text{(A4)} \\ \overline{\boldsymbol{y}}(\eta) = \boldsymbol{B}(t_f - \eta)^T \overline{\boldsymbol{x}}(\eta) & \text{(A5)} \end{cases}$$

with $\eta \in [0, t_f]$, $\overline{\boldsymbol{x}} \in \mathbb{R}^n$, $\overline{\boldsymbol{u}} \in \mathbb{R}^h$, and $\overline{\boldsymbol{y}} \in \mathbb{R}^k$. The system $\overline{\boldsymbol{\Sigma}}$ is sometimes also referred to as *adjoint model*.

The following important result links the system $\boldsymbol{\Sigma}$ and its dual $\overline{\boldsymbol{\Sigma}}$:

**Theorem A1.** *Given $\boldsymbol{K}$ and $\overline{\boldsymbol{K}}$, the impulse response matrices of $\boldsymbol{\Sigma}$ and $\overline{\boldsymbol{\Sigma}}$, respectively, the following relation holds:*

$$\boldsymbol{K}(t_f, \tau) = \overline{\boldsymbol{K}}(t_f - \tau, 0)^T \tag{A6}$$

The system response at $t_f$, therefore, can be written as:

$$\boldsymbol{y}(t_f) = \boldsymbol{C}(t_f)\boldsymbol{\Phi}(t_f, t_0)\boldsymbol{x}_0 + \int_{t_0}^{t_f} \overline{\boldsymbol{K}}(t_f - \tau, 0)^T \boldsymbol{u}(\tau)\mathrm{d}\tau \tag{A7}$$

Compared to Equation (A3), the computation of $\boldsymbol{y}(t_f)$ now requires only the knowledge of the impulse response of $\overline{\boldsymbol{\Sigma}}$, for impulses applied at time 0. Conversely, if Equation (A3) was used, the impulse response had to be computed at each integration step $\tau$. Therefore, from the computational point of view, Equation (A7) requires $h$ simulations of the dual system, one for each input $\overline{u}_i$.

It is worth noting that the impulse matrix $\overline{\boldsymbol{K}}(t_f - \tau, 0)^T$, or more specifically the element $\{\overline{\boldsymbol{K}}(t, 0)^T\}_{ij}$, provides a measurement of the influence of the $j$-the input applied at time $\tau$ on the $i$-th output at time $t_f$. Hence, the impulse matrix of the dual system can be used to understand the weight of the input on the system output, allowing for choosing the best combination of inputs able to achieve the desired output at time $t_f$. This also allows the study of the couplings of the system states, as shown in [27]. The impulse response of the dual system can be transformed into an initial value problem following the procedure described in [16,28].

The adjoint guidance is then applied to a two-impulse maneuver at each hold point. In other words, the chaser leaves a hold point with an impulsive firing and reaches the next with a braking impulse. Consider the impulsive control vector:

$$\boldsymbol{u}(t) = \boldsymbol{d}\delta(t - t_1), \quad \boldsymbol{d} \in \mathbb{R}^k \tag{A8}$$

where $\delta(t - t_1)$ is the unitary impulse applied at $t_1 \geq t_0$. We can compute the response of $\boldsymbol{\Sigma}$ to $\boldsymbol{u}(t)$ on the $i$-th output by means of the associate dual system as follows:

$$y_i(t_f) = \overline{\boldsymbol{x}}^i(t_f - t_0)^T \boldsymbol{x}_0 + \int_{t_0}^{t_f} \overline{\boldsymbol{y}}^i(t_f - \tau)^T \boldsymbol{d}\delta(\tau - t_1)\mathrm{d}\tau \tag{A9}$$

$$= \overline{\boldsymbol{x}}^i(t_f - t_0)^T \boldsymbol{x}_0 + \overline{\boldsymbol{y}}^i(t_f - t_1)^T \boldsymbol{d} \tag{A10}$$

Introducing the following matrices

$$\overline{\boldsymbol{X}}(t) = \mathrm{ver}\left\{\overline{\boldsymbol{x}}^i(t)^T\right\}_{i=1\ldots h} \in \mathbb{R}^{h \times n}, \quad \overline{\boldsymbol{Y}}(t) = \mathrm{ver}\left\{\overline{\boldsymbol{y}}^i(t)^T\right\}_{i=1\ldots h} \in \mathbb{R}^{h \times k} \tag{A11}$$

obtained by vertical concatenation through the operator $\mathrm{ver}\{\cdot\}$, the response of the system $\boldsymbol{\Sigma}$ is written as:

$$\boldsymbol{y}(t_f) = \overline{\boldsymbol{X}}(t_f - t_0)\boldsymbol{x}_0 + \overline{\boldsymbol{Y}}(t_f - t_1)\boldsymbol{d} \tag{A12}$$

Hence, the required impulse magnitude to obtain the output $\boldsymbol{y}(t_f)$ can be computed as follows:

$$\boldsymbol{d} = \overline{\boldsymbol{Y}}(t_f - t_1)^* \left(\boldsymbol{y}(t_f) - \overline{\boldsymbol{X}}(t_f - t_0)\boldsymbol{x}_0\right) \tag{A13}$$

where the operator $(\cdot)^*$ denotes the pseudo-inverse, whose computation is related to the degree of controllability of the system [29]. Starting from this result, we compute a two-impulse maneuver for the chaser, based on the ELERM in Equations (34) and (35).

In particular, if we consider the system output $\boldsymbol{y}$ as the whole state, then:

$$\boldsymbol{y}(t) = \boldsymbol{x}(t) = \begin{bmatrix} \boldsymbol{\rho}(t) \\ [\dot{\boldsymbol{\rho}}(t)]_{\mathcal{L}} \end{bmatrix} \tag{A14}$$

Two impulses are fired at $t_1$ and $t_2$, i.e., $\boldsymbol{u}_1(t) = \boldsymbol{d}_1\delta(t - t_1)$ and $\boldsymbol{u}_2(t) = \boldsymbol{d}_2\delta(t - t_2)$, with $t_0 \leq t_1 < t_2 \leq t_f$. The system response under a two-impulse maneuver is computed according to Equation (A12):

$$\begin{bmatrix} \boldsymbol{\rho}_f \\ \dot{\boldsymbol{\rho}}_f \end{bmatrix} = \begin{bmatrix} \overline{\boldsymbol{X}}_{\boldsymbol{\rho}}(t_f - t_0) \\ \overline{\boldsymbol{X}}_{\dot{\boldsymbol{\rho}}}(t_f - t_0) \end{bmatrix} \begin{bmatrix} \boldsymbol{\rho}_0 \\ \dot{\boldsymbol{\rho}}_0 \end{bmatrix} + \begin{bmatrix} \overline{\boldsymbol{Y}}_{\boldsymbol{\rho}}(t_f - t_1) \\ \overline{\boldsymbol{Y}}_{\dot{\boldsymbol{\rho}}}(t_f - t_1) \end{bmatrix} \boldsymbol{d}_1 + \begin{bmatrix} \overline{\boldsymbol{Y}}_{\boldsymbol{\rho}}(t_f - t_2) \\ \overline{\boldsymbol{Y}}_{\dot{\boldsymbol{\rho}}}(t_f - t_2) \end{bmatrix} \boldsymbol{d}_2 \tag{A15}$$

where $(\boldsymbol{\rho}_f, \dot{\boldsymbol{\rho}}_f)$ is the desired relative state, and $\overline{\boldsymbol{X}}$ and $\overline{\boldsymbol{Y}}$ are obtained by simulating the dual system associated with the ELERM. With $\overline{\boldsymbol{X}}_{\boldsymbol{\rho}}$ and $\overline{\boldsymbol{Y}}_{\boldsymbol{\rho}}$, we denoted the first three rows of $\overline{\boldsymbol{X}}$ and $\overline{\boldsymbol{Y}}$ associated with the relative position, whereas $\overline{\boldsymbol{X}}_{\dot{\boldsymbol{\rho}}}$ and $\overline{\boldsymbol{Y}}_{\dot{\boldsymbol{\rho}}}$ are the rows associated with the relative velocity vector.

The classical two-impulse maneuver is composed by a first impulse, which is used to target the desired final position, and by a second impulse to stop the chaser at the end.

Thus, according to Equation (A15), the first impulse magnitude is computed considering its effects on the final relative position:

$$\boldsymbol{\rho}_f = \overline{\boldsymbol{X}}_{\boldsymbol{\rho}}(t_f - t_0)\boldsymbol{\rho}_0 + \overline{\boldsymbol{Y}}_{\boldsymbol{\rho}}(t_f - t_1)\boldsymbol{d}_1 \tag{A16}$$

The first impulse magnitude is then:

$$\boldsymbol{d}_1 = \overline{\boldsymbol{Y}}_{\boldsymbol{\rho}}(t_f - t_1)^* \left( \boldsymbol{\rho}_f - \overline{\boldsymbol{X}}_{\boldsymbol{\rho}}(t_f - t_0)\boldsymbol{\rho}_0 \right) \tag{A17}$$

The braking maneuver to execute at $t_2$ is computed considering again Equation (A15), plus the influence of the first impulse on the chaser motion:

$$\dot{\boldsymbol{\rho}}_f = \overline{\boldsymbol{X}}_{\dot{\boldsymbol{\rho}}}(t_f - t_0)\dot{\boldsymbol{\rho}}_0 + \overline{\boldsymbol{Y}}_{\dot{\boldsymbol{\rho}}}(t_f - t_1)\boldsymbol{d}_1 + \overline{\boldsymbol{Y}}_{\dot{\boldsymbol{\rho}}}(t_f - t_2)\boldsymbol{d}_2 \tag{A18}$$

Thus:

$$\boldsymbol{d}_2 = \overline{\boldsymbol{Y}}_{\dot{\boldsymbol{\rho}}}(t_f - t_2)^* \left( \dot{\boldsymbol{\rho}}_f - \overline{\boldsymbol{X}}_{\dot{\boldsymbol{\rho}}}(t_f - t_0)\dot{\boldsymbol{\rho}}_0 - \overline{\boldsymbol{Y}}_{\dot{\boldsymbol{\rho}}}(t_f - t_1)\boldsymbol{d}_1 \right) \tag{A19}$$

Equations (A17) and (A19) consider the impulse firing along the three axes. Impulses along a preferred firing direction (e.g., radial or tangential impulses) can be easily derived.

The performance of the guidance was evaluated computing relative distance and velocity errors as:

$$e_{\boldsymbol{\rho}} = \|\boldsymbol{\rho}_f - \boldsymbol{\rho}(t_f)\|, \quad e_{\dot{\boldsymbol{\rho}}} = \|\dot{\boldsymbol{\rho}}_f - \dot{\boldsymbol{\rho}}(t_f)\| \tag{A20}$$

and total control effort:

$$\Delta v = \int_{t_0}^{t_f} \|\boldsymbol{u}(t)\| \mathrm{d}t \tag{A21}$$

*Appendix A.2. PID Control*

During the initial phase of the rendezvous, the primary concern is to follow an attitude profile, or to maintain the target in the field of view of the chaser's cameras. To this end, a standard PID controller on all axes was considered to be satisfactory. The controllers were simply tuned to obtain a good rise time.

An example of the performance without and with the controller are shown in Figure A1 and Figure A2, respectively. The simulation considers a nonzero initial condition on the angular velocity with no damping at the beginning of the maneuver. The continuous rotation of the vehicle can be seen in Figure A1. After the insertion of the controller, the vehicle is stabilized to the desired attitude (zero in this case) as shown in Figure A2. The presence of the PID also allows the pointing of the cameras always towards the target.

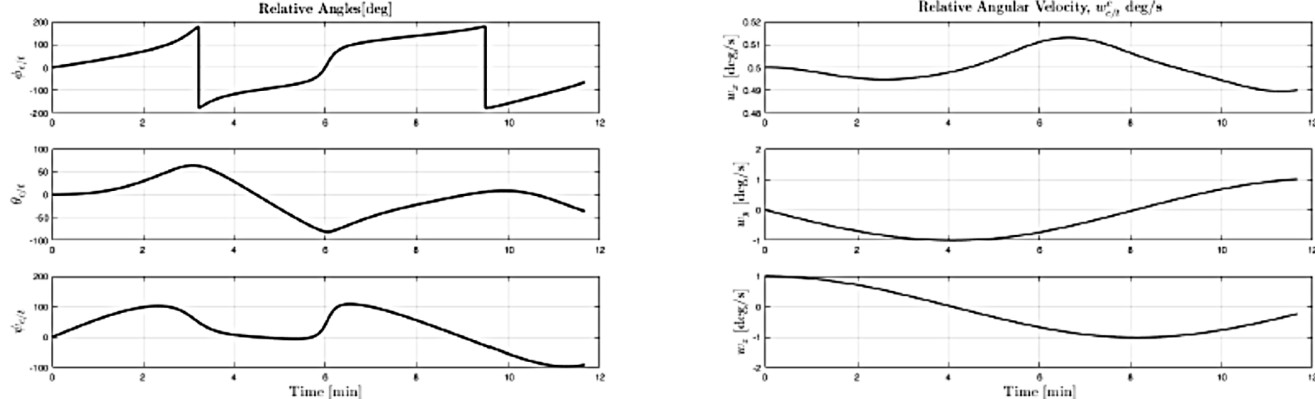

**Figure A1.** Attitude and attitude rate no PID.

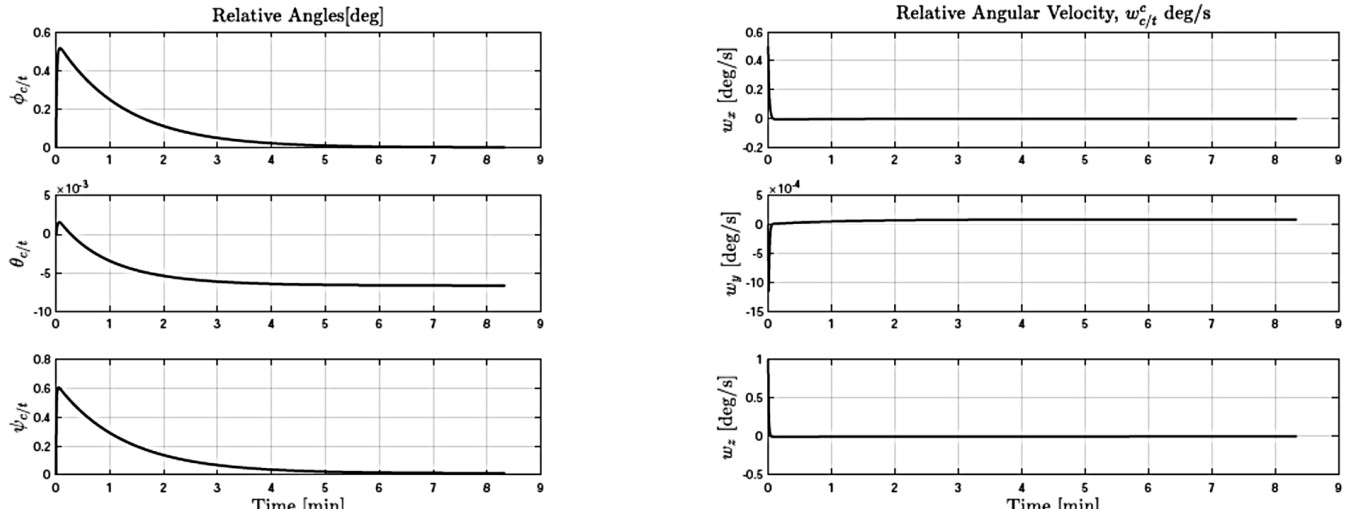

**Figure A2.** Attitude and attitude rate with PID.

*Appendix A.3. SDRE Guidance*

Since the middle of the 1990s, State-Dependent Riccati Equation (SDRE) strategies have emerged as general design methods that provide a systematic and effective means of designing nonlinear controllers, observers, and filters. These methods overcome many of the difficulties and shortcomings of existing methodologies, and deliver computationally simple algorithms that have been highly effective in a variety of practical and meaningful applications. In a special session at the 17th IFAC Symposium on Automatic Control in Aerospace 2007, theoreticians and practitioners in this area of research were brought together to discuss and present SDRE-based design methodologies as well as review the supporting theory. It became evident that the number of successful simulations, experimental, and practical real-world applications of SDRE control have outpaced the available theoretical results.

The methodology originated as an extension of Linear Quadratic Control (LQR) techniques to a special class of nonlinear systems. An interesting survey on the method can be found in [21]. Stability and controllability aspects can be found in [22,23]. SDRE controllers were used in space related applications, especially for the control of proximity operation and formation flight [24–26].

The method entails a factorization (that is, parameterization) of the nonlinear dynamics into a state vector and the product of a matrix-valued function that depends on

the state itself. In doing so, the SDRE algorithm fully captures the nonlinearities of the system, bringing the nonlinear system to a (non-unique) linear structure having state-dependent coefficient (SDC) matrices, and minimizing a nonlinear performance index having a quadratic-like structure. An algebraic Riccati equation (ARE) using the SDC matrices is then solved online to give the suboptimum control law.

Let us consider a nonlinear regulator problem for minimizing the cost function in Equation (A22):

$$J(\mathbf{x}, \mathbf{u}) = \frac{1}{2} \int_0^\infty \left( \mathbf{x}^\top \mathbf{Q}(\mathbf{x})\mathbf{x} + \mathbf{u}^\top \mathbf{R}(\mathbf{x})\mathbf{u} \right) dt \tag{A22}$$

subject to a nonlinear differential constraint and affine in control (A23):

$$\dot{\mathbf{x}} = \mathbf{f}(\mathbf{x}) + \mathbf{g}(\mathbf{x})\mathbf{u} \tag{A23}$$

where $\mathbf{x} \in \mathbb{R}^n$ is state vector, $\mathbf{u} \in \mathbb{R}^m$ is control vector, $\mathbf{f} : \mathbb{R}^n \to \mathbb{R}^n$, $\mathbf{g}(\mathbf{x}) \neq \mathbf{0} \ \forall \mathbf{x} \in \mathbb{R}^n$. $\mathbf{Q}(\mathbf{x}) \geq 0$ and $\mathbf{R}(\mathbf{x}) > 0$ are the weight matrices of the state vector and the input vector, respectively. If the dynamics of the system can be written in a pseudo-linear form (A24) by means of *State Dependent Coefficient* (SDC) matrices parametrization:

$$\dot{\mathbf{x}} = \mathbf{A}(\mathbf{x})\mathbf{x} + \mathbf{B}(\mathbf{x})\mathbf{u} \tag{A24}$$

Note that the SDC parametrization is not unique and provides the designer with additional degrees of freedom to enhance controller performance. The SDRE control method can be summarized in the following two steps, both occurring at each integration time. First, the state dependent ARE is solved, then the nonlinear full state (in the most simplest case) feedback control law is found as shown in Equation (A25):

$$\mathbf{u} = -\mathbf{R}^{-1}(\mathbf{x})\mathbf{B}^\top(\mathbf{x})\mathbf{P}(\mathbf{x})\mathbf{x} \tag{A25}$$

Rendezvous maneuvers can also be executed imposing constraints on relative position and velocity, especially at close distances, where relative motion precision is necessary. The SDRE structure can be used to incorporate such constraints.

Consider a system given by Equation (A23), with $\mathbf{x}(0) = \mathbf{x}_0 \in \Omega$ and the set of admissible states defined by Equation (A26):

$$\Omega = \left\{ \mathbf{x} : \mathbf{l}(\mathbf{x}) \leq \mathbf{0}, \mathbf{l}(\mathbf{x}) \in \mathbb{R}^p, \mathbf{l}(\cdot) \in C^1 \right\} \tag{A26}$$

It is possible to synthesize a SDRE controller that stabilizes the closed loop system, and, like $\mathbf{x}$, does not go beyond $\partial\Omega$, the border of $\Omega$, defined as:

$$\partial\Omega = \left\{ \mathbf{x} : \mathbf{l}(\mathbf{x}) = \mathbf{0}, \mathbf{l}(\mathbf{x}) \in \mathbb{R}^p, \mathbf{l}(\cdot) \in C^1 \right\} \tag{A27}$$

A sufficient condition for $\mathbf{x}$ to remain within $\Omega$ is $\dot{\mathbf{l}}(\mathbf{x}) = \mathbf{0}$, see [30,31]:

$$\nabla\mathbf{l}(\mathbf{x})\dot{\mathbf{x}} = \nabla\mathbf{l}(\mathbf{x})\left[\mathbf{f}(\mathbf{x}) + \mathbf{g}(\mathbf{x})\mathbf{u}\right] = \mathbf{0} \tag{A28}$$

We can parametrize the above condition in a SDC form by adding a fictitious output $\mathbf{z}$ to the problem:

$$\begin{aligned} \mathbf{z} &= \nabla\mathbf{l}(\mathbf{x})\left[\mathbf{A}(\mathbf{x})\mathbf{x} + \mathbf{B}(\mathbf{x})\mathbf{u}\right] \\ &= \mathbf{C}(\mathbf{x})\mathbf{x} + \mathbf{D}(\mathbf{x})\mathbf{u} \end{aligned} \tag{A29}$$

Such controller forces the closed loop trajectories to follow the level curves of set $\Omega$. Taking into account the constraint, the cost function changes to Equation (A30) :

$$J(\mathbf{x}, \mathbf{u}) = J_0(\mathbf{x}, \mathbf{u}) + J_\Omega(\mathbf{x}, \mathbf{u}) =$$
$$= \frac{1}{2} \int_0^\infty \left( \mathbf{x}^T \mathbf{Q}(\mathbf{x}) \mathbf{x} + \mathbf{u}^T \mathbf{R}(\mathbf{x}) \mathbf{u} \right) dt + \frac{1}{2} \int_0^\infty \left( \mathbf{z}^T \mathbf{W}_z(\mathbf{x}) \mathbf{z} \right) dt \quad \text{(A30)}$$

where $\mathbf{W}_z$ is a $p \times p$ weight matrix, such that its *i*-th element has a large value when $\mathbf{x}$ is near the border of the *i*-th constraint, and small otherwise. This implies that the component $J_\Omega(\mathbf{x}, \mathbf{u})$ in the cost function dominates with respect to $J_0(\mathbf{x}, \mathbf{u})$ when the state vector does not satisfy the constraint, and becomes negligible when the constraint is satisfied. The matrices in the Riccati equation are now modified as follows:

$$\bar{\mathbf{A}}^T(\mathbf{x})\bar{\mathbf{P}}(\mathbf{x}) + \bar{\mathbf{P}}(\mathbf{x})\bar{\mathbf{A}}(\mathbf{x}) - \bar{\mathbf{P}}(\mathbf{x})\bar{\mathbf{B}}(\mathbf{x})\bar{\mathbf{R}}^{-1}(\mathbf{x})\bar{\mathbf{B}}^T(\mathbf{x})\bar{\mathbf{P}}(\mathbf{x}) + \bar{\mathbf{Q}}(\mathbf{x}) = 0,$$

$$\bar{\mathbf{R}}(\mathbf{x}) = \mathbf{R}(\mathbf{x}) + \mathbf{D}^T(\mathbf{x})\mathbf{W}_z(\mathbf{x})\mathbf{D}(\mathbf{x}),$$

$$\text{(A31)}$$

$$\bar{\mathbf{Q}}(\mathbf{x}) = \mathbf{Q}(\mathbf{x}) + \mathbf{C}^T(\mathbf{x})\mathbf{W}_z(\mathbf{x})\left[\mathbf{I} - \mathbf{D}^T(\mathbf{x})\bar{\mathbf{R}}^{-1}(\mathbf{x})\mathbf{D}(\mathbf{x})\right]\mathbf{W}_z(\mathbf{x})\mathbf{C}(\mathbf{x})$$

$$\bar{\mathbf{A}}(\mathbf{x}) = \mathbf{A}(\mathbf{x}) - \mathbf{B}(\mathbf{x})\bar{\mathbf{R}}^{-1}(\mathbf{x})\mathbf{D}^T(\mathbf{x})\mathbf{W}_z(\mathbf{x})\mathbf{C}(\mathbf{x}).$$

The resulting control law becomes:

$$\mathbf{u} = -\mathbf{K}(\mathbf{x})\mathbf{x}$$
$$= -\left[\mathbf{K}_0(\mathbf{x}) + \mathbf{K}_\Omega(\mathbf{x})\right]\mathbf{x} \quad \text{(A32)}$$

where

$$\mathbf{K}_0(\mathbf{x}) = \bar{\mathbf{R}}^{-1}(\mathbf{x})\mathbf{B}(\mathbf{x})\bar{\mathbf{P}}(\mathbf{x})$$
$$\mathbf{K}_\Omega(\mathbf{x}) = \bar{\mathbf{R}}^{-1}(\mathbf{x})\mathbf{D}^T(\mathbf{x})\mathbf{W}_z(\mathbf{x})\mathbf{C}(\mathbf{x}) \quad \text{(A33)}$$

$\mathbf{K}_0(\mathbf{x})$ is the controller component responsible for stability and performance, while $\mathbf{K}_\Omega(\mathbf{x})$ satisfies the constraints [31].

There are situations where $\nabla l(\mathbf{x})$ is orthogonal to $\mathbf{B}(\mathbf{x})$, so that $\mathbf{D}(\mathbf{x}) = \mathbf{0}$ e $\bar{\mathbf{R}}(\mathbf{x}) = \mathbf{R}(\mathbf{x})$. In this case, a possible choice for $\mathbf{W}_z(\mathbf{x})$ is that the *i*-th element be selected to be large, when we are in a region of the state space to be penalized, and zero otherwise [30].

As an example, consider the graphical proximity representation in Figure A3, in which $\mathbf{p}$ is the unit direction vector of path direction and $\beta$ is the maximum cone angle of the desired final corridor. In order to satisfy this constraint, we rewrite it using (A26) as follows:

$$l(\mathbf{x}) = -\left[\boldsymbol{\rho}^\top\right]_\mathcal{T} \cdot \left[\mathbf{p}\right]_\mathcal{T} + ||\boldsymbol{\rho}||_\mathcal{T} \cos \beta \leq 0 \quad \text{(A34)}$$

where $\mathbf{x} = \left[\boldsymbol{\rho}^\top, \dot{\boldsymbol{\rho}}^\top, \mathbf{q}_{c/l}^\top, \boldsymbol{\omega}_{c/l}^\top\right]^\top$. It is important to note that the constraint given by Equation (A34) is expressed in $\mathcal{T}$ frame, so, as we can see from Equation (A34), the direction of cone axis depends on target's attitude. From Equation (A29), we obtain the fictitious output $\mathbf{z}$.

Note that, in our case, $\nabla l(\mathbf{x}) \perp \mathbf{B}(\mathbf{x})$, so the weight function $\mathbf{W}_z$ was selected to penalize the state when it is far from the imposed constraint. To this end, the weight function depends on the 3D distance of chaser's Center of mass (CoM) from the line described by the unit vector $\mathbf{p}$. In this way, $\mathbf{W}_z$ has a large value when the chaser's CoM is far from the cone axis and a small value when it is close to the axis.

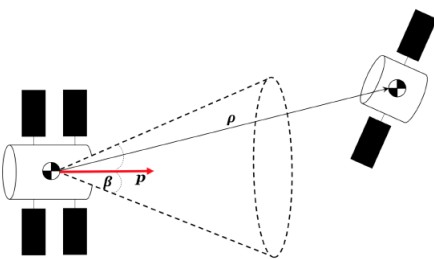

**Figure A3.** Proximity example scenario.

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
