# Peer review of "Rendezvous in Cis-Lunar Space near Rectilinear Halo Orbit: Dynamics and Control Issues"

_aerospace, doi:10.3390/aerospace8030068_

Round 1

Reviewer 1 Report

The article treats a demanding problem and presents valuable results.

There are a few issues, which should be clarified:

  • The Earth-Moon system is quite strongly influenced by the sun's gravity. If that is neglected, the three-body problem simplifies quite a lot, only the (quite complicated) dynamics of the rocket has to be found. When taking the sun into account, already the Sun-Earth-Moon system can be solved only numerically. It should be stated clearly, which model was used.
  • The authors consider the minimization of fuel expenditure, which in fact leads to a L1-problem, which cannot be solved as easily as quadratic problems. Especially in some cases (the author's do not present the full set of control laws; so I am not sure that it applies here) it is impossible to find all control values. The authors should explain, how the L1-problem is resolved.
  • The treatment of inequality constraints for state variables looks also questionable: There are different kinds of such constraints possible, depending on how often the constraint equation has to be differentiated until an equation for the control variables is obtained. For constraints l(x) of second order (one has to calculate l''(x) to obtain constraints on the control) jumps in the adjoint variables could occur.
    By the presented method the quantity l(x) would remain constant all the time, which might lead to a sub-optimal control.
  • The authors use the pseudo-inverse to calculate the control magnitudes di. If the matrix is singular, the method would yield good candidates di, but the boundary conditions wouldn't be satisfied. For a regular matrix there is no need for the pseudo-inverse.
  • At some instances the authors mention Euler angles and quaternions. Since the Euler angles can become singular, quaternions would be safer. Are Euler angles used at all? Since the authors talk of (3,2,1), it seems, that they actually use Bryant angles, for Euler angles the first and last rotations are usually performed about the same axis. The notations for the rotation matrices could be shortened, by stating the rotation matrices R_i(theta_j) about axis i.
  • Figure 20 seems to be rotated by 180°. In (26) it isn't clear, whether it should read (c2/(1000 toff)) or (c2/1000) toff. (A TeXnical remark: functions like exp should be written upright (\exp), also indices like 'on' or 'off' should be written upright.) The "e" in "ton e toff" should be replaced by "and".
  • Formula (23) needs to be reformatted. Also the accents appear at the wrong places.
  • The last box in Fig. 14 is hard to read.
  • In Fig. 4a the direction of the vectors is hard to understand.
  • Shouldn't μ be denoted as Moon/Earth mass ratio?
  • The authors frequently mention "Local vertical local Horizon" frames; I think it should read "horizontal".
  • The article needs an extensive editing of English language and style. There are quite a number of grammatical mistakes; mainly singular and plural forms are mixed up. But there are also incomplete sentences, like "With state variable(s) defined from (by) Eq. 7" or "Where Bs is called ...".

Author Response

  • The structure of the manuscript has been improved with the addition of the introduction and extended conclusions.  An appendix has been added. English and typos corrected.
  • The influence of the Sun is negligible at this preliminary stage (explained)
  • The optimization procedure yields a suboptimal solution. the constraint l(x) is assumed sufficiently smooth (see appendix A3).
  • The computation of the pseudoinverse in (49) is related to the degree of controllability as described in the added reference (29)
  • Fig 20 and eq "3) corrected.
  • Fig 14 corrected.
  • Fig 4a seems clear.
  • mu can be named in different ways.
  •  LVLH has the standard definition as in the manuscript.

Reviewer 2 Report

The article is a very pertinent and the Authors have developed a remarkably interesting topic, full of ideas. Despite this, some integration and clarification are needed: for these reasons I have proposed a "minor revision".

Some comments:

  1. Section 1 “Background”: it would be advisable for the authors to insert a paragraph on the "state of the art" or a small introduction on similar missions in which Lagrangian points were used as fire of orbits of satellites, such as the mission involving the Queqiao sat. which is already positioned in the Lagrangian point L2
  2. Section 7 “Conclusion”: this section is also too concise and schematic: please better focus the conclusions by highlighting the innovative part

Minor comments:

  1. Figure 3 (page 3) is too little and not clear: please improve dimension and resolution
  2. Figure 5 (page 5) is too little and not clear: please improve dimension and resolution
  3. Figures 6/a/b/c (page 8) are too little and not clear: please improve dimensions and resolution
  4. Figures 7/a/b/c/d (page 9) are too little and not clear: please improve dimensions and resolution
  5. Figures 8/a/b/c/d (page 10) are too little and not clear: please improve dimensions and resolution
  6. Figures 9/a/b/c (page 11) are too little and not clear: please improve dimensions and resolution
  7. Figure 10 (page12) is too little and not clear: please improve dimension and resolution
  8. Figure 11 (page12) is too little and not clear: please improve dimension and resolution
  9. Figures 12/a/b/c/d (page 13) are too little and not clear: please improve dimensions and resolution
  10. Figures 13/a/b/c/d (page 13) are too little and not clear: please improve dimensions and resolution
  11. Figure 14 (page 14) is too little and not clear: please improve dimension and resolution
  12. Figures 21/a/b (page 20) are too little and not clear: please improve dimensions and resolution
  13. Figures 22/a/b (page 21) are too little and not clear: please improve dimensions and resolution
  14. Figure 24 (page 25) is too little and not clear: please improve dimension and resolution
  15. Figure 25 (page 25) is too little and not clear: please improve dimension and resolution
  16. Figure 27 (page 28) is too little and not clear: please improve dimension and resolution
  17. Figure 29 (page 30) is too little and not clear: please improve dimension and resolution
  18. Figure 30 (page 30) is too little and not clear: please improve dimension and resolution
  19. Figure 31 (page 30) is too little and not clear: please improve dimension and resolution
  20. Figure 32 (page 31) is too little and not clear: please improve dimension and resolution
  21. It is recommended an extensive reading to correct some sentences and typo errors.

Author Response

-The manuscript has an introduction and extended conclusions, which better describe the work.

- the figures have been enlarged and clarified.

Reviewer 3 Report

This paper covers multiple aspects of spacecraft dynamics and control, viz. orbital dynamics, attitude dynamics, sensor and actuator selection, and guidance, with the ultimate purpose of designing a rendezvous mission. The rendezvous is between an unmanned lunar ascent vehicle carrying samples of lunar soil, and an orbiting ground station containing astronauts. I appreciate the topic and the effort put forth by the authors to navigate though and integrate various sub-disciplines for the overall mission. While the work is in a good direction, I believe that a lot of improvement is still necessary. 

1. The novelty of this paper is not adequately highlighted. The Introduction section should clearly say with appropriate references which part of the work has been attempted before, which research gaps the current work seeks to fill and how. The Introduction also needs to discuss the organization of the rest of the paper.

2. The orientations are described using Euler angles which have singularities. The authors need to explain how later in the paper they plan to design the maneuver such that the singularity will never arise, i.e. the middle angle will never become a certain number (0 or 90 deg). At the same time, what is the difficulty to use quaternions or Modified Rodriguez Parameters (MRPs) to describe orientation for the maneuver in the Results section? Additionally, Figure 6 shows jumps in the pitch and yaw attitude angles. What is the reason behind these jumps, and can we afford to have them for this mission? Physically, what kind of tumbling motion do these jumps mean? 

3. Tables 1 - 3 show errors due to using the newly derived equations of motion against a standard framework. While I appreciate the effort as a means to check for numerical issues in simulation, it is not clear how a reader not involved in space research will interpret these numbers: are they good or bad? Some numbers are as large as 10,000 km, while some are as small as 10^(-5) km/s. The authors need to discuss some criteria for acceptable errors. 

4. Some acronyms seem to appear 'suddenly' as one reads the paper. Acronyms such as LOP-G, RCS, ELERM, CLERM, etc. are used with no mention of the full forms. For some others such as ER3BP, CR3BP, WAC, NAC, etc. the reader has to guess the full form from previous lines of text. I would urge the authors to make a separate table of all acronyms with their full forms. This is a necessary step to enhance the paper's readability to people outside of the domain of spacecraft dynamics and control. 

5. Equation (20) has symbols such as ang_{px} and alpha_1 which were not defined until after equation (21). If possible, the authors may consider making a separate list of frequently used symbols to avoid such issues.   

6. Section 5 discusses Adjoint Guidance, PID, and SDRE in detail. The novelty is not the development of a particular guidance algorithm, but the application of all three existing algorithms depending on the stage of the mission. Is it really necessary to include all theoretical details of an algorithm, or can the authors focus more on why a specific algorithm is suitable for a specific part of the mission? Most likely it is enough to just mention a few good references for the theoretical details. Alternatively, the details can be moved to an Appendix. 

My overall assessment is that the work is in a good direction, but the technical content and the presentation need some major work. I recommend that the authors make a major revision and resubmit. 

Author Response

  • The English has been revised and typos edited.
  • The manuscript now has an introduction that presents the work and defines the original contributions. The conclusions section is expanded.
  • The details of the guidance solutions are now collected in the appendix, to better the flow of the manuscript.
  • The Euler/quaternion issue has been explained see also eqs 39-41.
  • An abbreviations list is added at the beginning.
  • The amount of errors in the propagation equations and the relative errors are explained. An additional table was added to provide requirements on safety issues for more detailed analysis, which is outside the scope of the work.

Round 2

Reviewer 1 Report

The answers are not really satisfactory.

The question regarding the L1 optimization hasn't been answered at all.

The directions in Fig. 4a can be concluded from the text but not from the figure.

It makes a difference for me, whether it is the earth/moon or moon/earth mass ratio. The question about the constraints had nothing to do with smothness, but with the kind of the constraint.

The authors should answer all stated questions.

Author Response

  1. the proble of L1 optimization has been resolved. Our mistake was the typo in eq 33, which was overseen by us. There is no quadratic optimization and we have corrected the equation with hopefully a clear explanation.
  2. Regardin figure 4a, the LVLH frame has been further clarified. The figure itself is qualitative and used in many publications. If the reviewer feels the figure not clear we could erase it. It is true that V-bar, H-bar, and R-bar are formally defined for Keplerian orbits (mostly circular), but the "lingo" is maintained in the community.
  3. The mass ratio mu has been clarified. At first reading the reviewer was correct, in that it is the moon/earth ratio. This has been explained.

We hope that these corrections make the manuscript more correct, and we thank the reviewers for the helpful comments.

Sincerely,

GIordana Buccioni, Mario Innocenti